# An evolutionary switch in ND2 enables Src kinase regulation of NMDA receptors

David P. Scanlon[1,2,*], Alaji Bah[3,4,*], Mickaël Krzeminski[3,4], Wenbo Zhang[1,2], Heather L. Leduc-Pessah[1,2], Yi Na Dong[1,2], Julie D. Forman-Kay[3,4] & Michael W. Salter[1,2]

The non-receptor tyrosine kinase Src is a key signalling hub for upregulating the function of N-methyl D-aspartate receptors (NMDARs). Src is anchored within the NMDAR complex via NADH dehydrogenase subunit 2 (ND2), a mitochondrially encoded adaptor protein. The interacting regions between Src and ND2 have been broadly identified, but the interaction between ND2 and the NMDAR has remained elusive. Here we generate a homology model of ND2 and dock it onto the NMDAR via the transmembrane domain of GluN1. This interaction is enabled by the evolutionary loss of three helices in bilaterian ND2 proteins compared to their ancestral homologues. We experimentally validate our model and demonstrate that blocking this interaction with an ND2 fragment identified in our experimental studies prevents Src-mediated upregulation of NMDAR currents in neurons. Our findings establish the mode of interaction between an NMDAR accessory protein with one of the core subunits of the receptor.

[1] Program in Neurosciences & Mental Health, The Hospital for Sick Children, 686 Bay St, Toronto, Ontario, Canada M5G 0A4. [2] Department of Physiology, University of Toronto, 1 King's College Circle, Toronto, Ontario, Canada M5S 1A8. [3] Program in Molecular Medicine, The Hospital for Sick Children, 686 Bay St, Toronto, Ontario, Canada M5G 0A4. [4] Department of Biochemistry, University of Toronto, 1 King's College Circle, Toronto, Ontario, Canada M5S 1A8. * These authors contributed equally to this work. Correspondence and requests for materials should be addressed to J.D.F.-K. (email: forman@sickkids.ca) or to M.W.S. (email: michael.salter@sickkids.ca).

Nmethyl-D-aspartate receptors (NMDARs)[6,7].

NADH dehydrogenase subunit 2 (ND2) is a mitochondrially encoded core subunit of complex I (NADH-quinone oxidoreductase)[1]. Mammalian complex I is composed of ~44 subunits[2,3], 7 of which are mitochondrially encoded[4], while the bacterial 'core' complex comprises 14 evolutionarily conserved subunits[5], including the bacterial homologue of ND2, NuoN. Additional to its role in complex I, an extra-mitochondrial role for ND2 has been defined in regulating N-methyl-D-aspartate receptors (NMDARs)[6,7].

NMDARs are a class of ionotropic glutamate receptors in the central nervous system (CNS)[8–10]. Physiological function of NMDARs is critical in a diversity of fundamental processes while NMDAR dysfunction is implicated in a diversity of pathological processes[11–13]. NMDARs function as large multi-protein complexes, centred on a core heterotetramer comprising two obligate GluN1 subunits and two subunits from the GluN2 family, GluN2A-D. The heterotetramer contains ligand-binding sites and an ion channel pore. The channel is gated open by binding of glycine or D-serine to each GluN1 subunit and glutamate to each GluN2 subunit. NMDARs have a four-layered modular domain architecture; an extracellular amino-terminal domain (ATD) and ligand-binding domain (LBD) are followed by a transmembrane domain (TMD) before ending with ~100–650 residue intrinsically disordered intracellular carboxy-terminal domain (CTD)[14,15]. Because of their amino-acid compositions, intrinsically disordered protein regions, such as the GluN2 CTD, lack stable secondary and tertiary structure, yet are increasingly recognized for critical biological roles they play in mediating regulatory protein interactions, particularly those involving post-translational modifications such as phosphorylation[16,17]. High-resolution structures of GluN1 and GluN2B heterotetrameric assemblies have revealed alternating GluN1–GluN2–GluN1–GluN2 subunits with a twofold symmetry between two GluN1–GluN2B heterodimers in the ATD and LBD, but with pseudo fourfold symmetry in the TMD[18]. The heterotetramer associates with multiple scaffolding, adapter and regulatory proteins that modulate NMDAR function, trafficking and localization[8,14].

Tyrosine phosphorylation of the NMDAR disordered CTD[19] is a key regulatory process that is implicated in a range of physiological functions and pathologies[20–22]. Src, the first tyrosine kinase found to upregulate NMDAR function[23], was subsequently shown to be key for hippocampal long-term potentiation[24,25]. Subsequently, Src emerged as a crucial hub through which multiple intracellular signalling cascades converge on NMDARs[21]. Src is a modular protein consisting of a C-terminal catalytic (Src homology 1 or SH1) domain, SH2 and SH3 interaction domains, a disordered region referred to as the unique domain (UD), and an N-terminal SH4 domain[26]. Anchoring of the kinase to NMDAR complexes[23] is essential for Src to phosphorylate GluN2 subunit CTDs and thereby upregulate channel activity[6]. We have previously shown that ND2 anchors Src to NMDARs at post-synaptic densities (PSDs) in the hippocampus[6], via an interaction involving Src UD residues 40–49 and ND2 residues 239–321 (ref. 7). Disrupting this interaction, with peptide fragments of Src or ND2, causes Src to disassociate from NMDARs while ND2 remains bound to the complex. This dissociation prevents Src from upregulating NMDAR channel activity, and thereby reduces pain hypersensitivity[7].

While ND2–Src interacting regions have been identified, a major unresolved issue is the mechanism by which ND2 itself is anchored to NMDAR complexes. All previously described NMDAR interacting proteins are known to associate with either the cytosolic CTD[27,28] or extracellular ATD regions[29–31]. However, based on amino-acid sequence analysis of ND2

homologues and recent structures of complex I (ref. 3), ND2 is a highly hydrophobic membrane protein comprised almost entirely of membrane-spanning helices. Therefore, contrary to the paradigm for NMDAR-interacting proteins, we postulated that ND2–NMDAR interaction might be mediated by ND2 binding to the NMDAR TMD.

Here we tested this possibility by generating a homology model of human ND2 and subsequently a docked model of this ND2 with the core NMDAR. This model demonstrates that the GluN1 TMD (GluN1–M4) can fit into a TM groove of ND2, but a similar interaction between ND2 and GluN2–M4 is precluded due to steric hindrance from GluN2 extracellular LBD. To validate this docked model, we used co-localization and bimolecular fluorescence complementation (BiFC) experiments to characterize association between ND2 and the core NMDAR, and to define the ND2 interacting region. Our results show that GluN1–M4, but not GluN2–M4, is indeed the critical NMDAR region for interacting with ND2, confirming our model. We also found that expressing the ND2-TM-6-8 region identified by our co-localization experiments prevents Src-mediated enhancement of NMDAR currents in hippocampal neurons by blocking ND2–Src association with the receptor. Thus, ND2–GluN1 interaction is essential for Src upregulation of NMDARs, which is critical in CNS physiology and pathophysiology. Finally, by comparing the evolutionary profiles of ND2 and NMDARs, we can begin to elucidate the key structural and functional features resulting in ND2-mediated Src upregulation of NMDARs.

## Results

**ND2 homology model reveals a novel interacting region.** To elucidate the basis for the ND2–NMDAR interaction, we required an atomic-level structure of ND2. As there is only a low-resolution cyro-electron microscopy (cryo-EM) model of mammalian ND2, in bovine mitochondrial complex I (ref. 3), we generated a homology model of human ND2 (Fig. 1), using crystal structures of ND2 homologues (NuoN) from *Escherichia coli* and *Thermus thermophilus* as templates[1,2]. According to our model, human ND2 is a single domain integral membrane protein consisting of 11 TM helices with a deep groove surrounded by TM helices 1, 5, 8 and 11 (Fig. 1a) that is a potential interacting surface. Based on our model, human ND2 has very limited extra-membranous segments to interact with the core NMDAR and an unusual surface envelope that strongly suggests a TMD interaction partner. This implies that the ND2–NMDAR interaction is likely mediated by the TMD of the NMDAR core.

Structure and sequence[1] alignments reveal that both bacterial templates used in generating our model contain three additional N-terminal helices relative to human ND2. In particular, the observed groove in the 'short' human ND2 results from the absence of the third TM helix of its bacterial homologue (Fig. 1a,b and Supplementary Fig. 1). Previous bioinformatics analysis has revealed that all pre-bilaterian organisms contain a 'long' ND2 sequence with three extra N-terminal TM helices, whereas all higher, bilaterian organisms only contain the 'short' ND2 sequence[32,33] (Supplementary Fig. 2). The groove is blocked in these lower organisms, which have the 'long' ND2 variant (Fig. 1b).

**NMDAR–ND2 model uncovers a novel TMD-based interaction.** We have previously shown that ND2 and NMDARs interact[6], but the exact nature of the interaction has remained elusive. To determine whether a direct interaction is structurally possible we performed *in silico* docking of our ND2 model onto the crystal

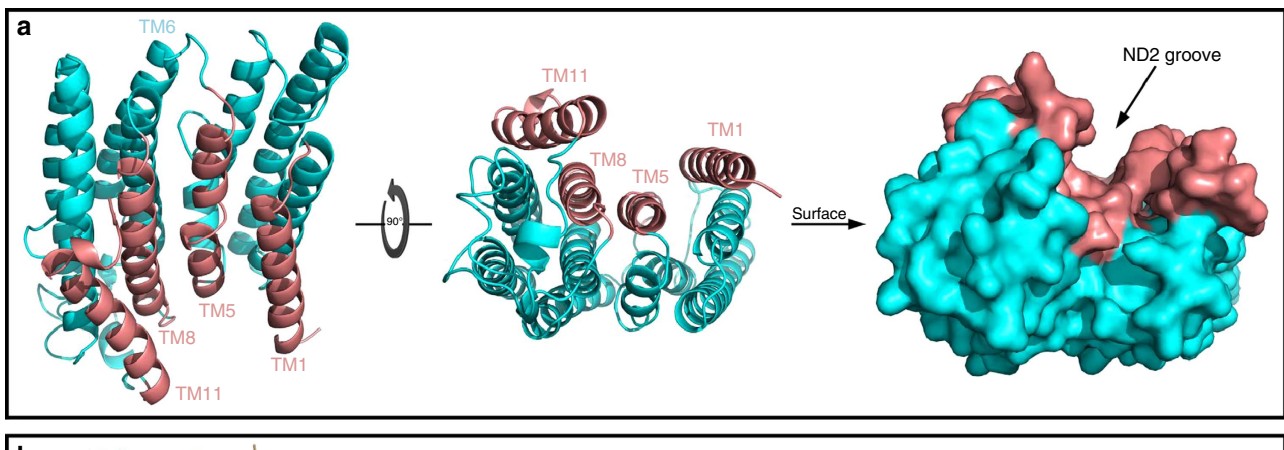

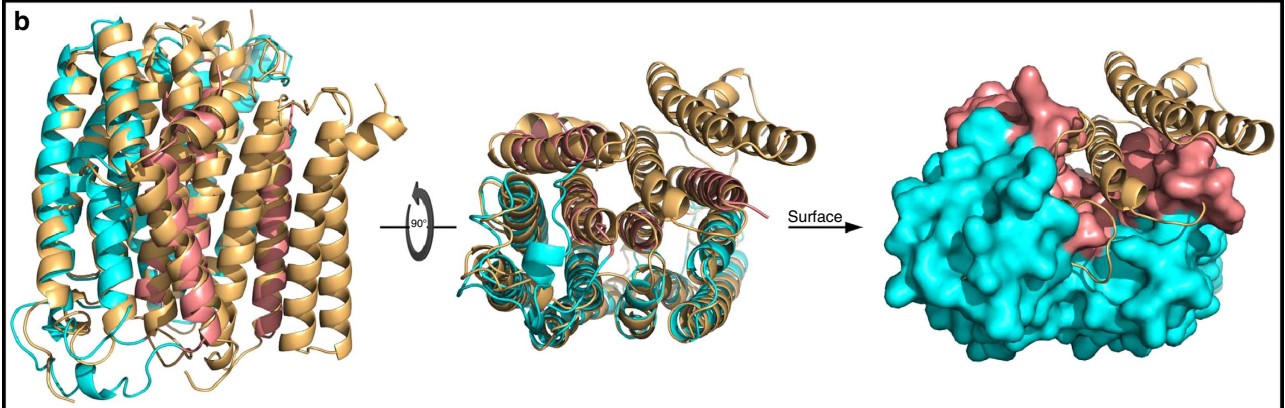

**Figure 1 | Model of the TMD-mediated interactions of the NMDAR–ND2 complex.** (**a**) Human ND2 homology model. Lateral and cytoplasmic views of the ND2 in cartoon and surface representations. The outer surface of ND2 is almost featureless on all sides except for the presence of a deep TM groove surrounded by TM helices 1, 5, 8 and 11 TMs, shown in salmon colour. (**b**) Structural basis for the formation of the ND2 TM groove. Structural alignment of the human ND2 homology model (cyan and salmon, as in **a**) with *E. coli.* NuoN subunit from complex I (beige, PDB code: 3RKO) in cartoon and surface representations. The evolutionary switch from 'long' to 'short' ND2 with the emergence of bilateral metazoans causes the loss of three N-terminal helices. The absence of the third helix results in the formation of a deep groove in 'short' ND2, including in the human ND2. See also Supplementary Figs 1 and 2.

structure of the NMDAR core complex[14,15] (Figs 1 and 2). According to these structures, each TMD subunit consists of four helices: three TM helices (M1, M3 and M4) and one central pore-like helix (M2) (Fig. 2a,b). Helices M1, M2 and M3 form the ion channel, while the exterior M4 connects the LBD and CTD and interacts predominantly with M1 and M3 of the neighbouring subunit. Consequently, each GluN M4 protrudes and exposes a significant amount of its surface, with two flanking shallow grooves formed between M1 and M4 of one subunit and between M4s of adjacent subunits (Fig. 2). We performed structural analysis using PyMol to investigate geometric and physico-chemical complementarities between the two structures before the computational docking simulation. For a direct ND2–TMD interaction, the protruding GluN M4 seemed a strong candidate compared to M1 to fit in the ND2 groove. The other NMDAR and ND2 surfaces did not show any potential complementary interacting regions. Therefore, we docked ND2 onto the NMDAR TMD with M4 inserted into the ND2 groove. Of the two possible cases, only the ND2–GluN1–M4 interaction was structurally feasible due to significant steric clashes between extracellular regions of ND2 and the LBD of GluN2, which approaches the membrane in both NMDAR crystal structures[14,15] (Fig. 2a).

Our docked model (Fig. 2c) shows extensive structural contacts between the two proteins (contact surface area ∼2,130 Å²), primarily involving ND2 interacting with GluN1–M4. GluN1–M4 fits compactly into the deep ND2 groove while TM1 and TM11 of ND2 interact with shallower grooves surrounding M4. As a result

of the GluN1–M4 interaction with M1 and M3 on neighbouring GluN2, our docked model places the Src-anchoring region of ND2 in close proximity to the GluN2 CTD, which is known to be phosphorylated at multiple tyrosine residues by Src[21]. The Src-anchoring region of ND2 was previously defined as containing residues 239–321, corresponding in our ND2 model to TM9-extracellular loop–TM10-intracellular loop. This final cytoplasmic element is, therefore, the likely Src-interacting region, based on our structural modeling, and could optimally position Src for targeting the GluN2 CTD (see below). Furthermore, according to our docked model, each GluN1 subunit can potentially interact with one ND2, allowing the NMDAR complex to include one or two ND2s.

**ND2 interacts with NMDARs.** To validate our docking model for the interaction between ND2 and the core NMDAR TMD, we developed an assay to monitor co-localization of ND2 with putative interaction partners. We co-transfected green fluorescent protein (GFP)-tagged ND2 and NMDAR subunits into human embryonic kidney cell line (HEK293) cells (Fig. 3 and Supplementary Fig. 3). GFP-ND2 transfected cells consistently displayed GFP fluorescence, and only fluorescing cells were selected for analysis. NMDAR subunits were monitored by indirect immunofluorescence and we quantified co-localization with ND2 by calculating thresholded Pearson's correlation coefficient (PCC) values[34]. We found that ND2 significantly co-

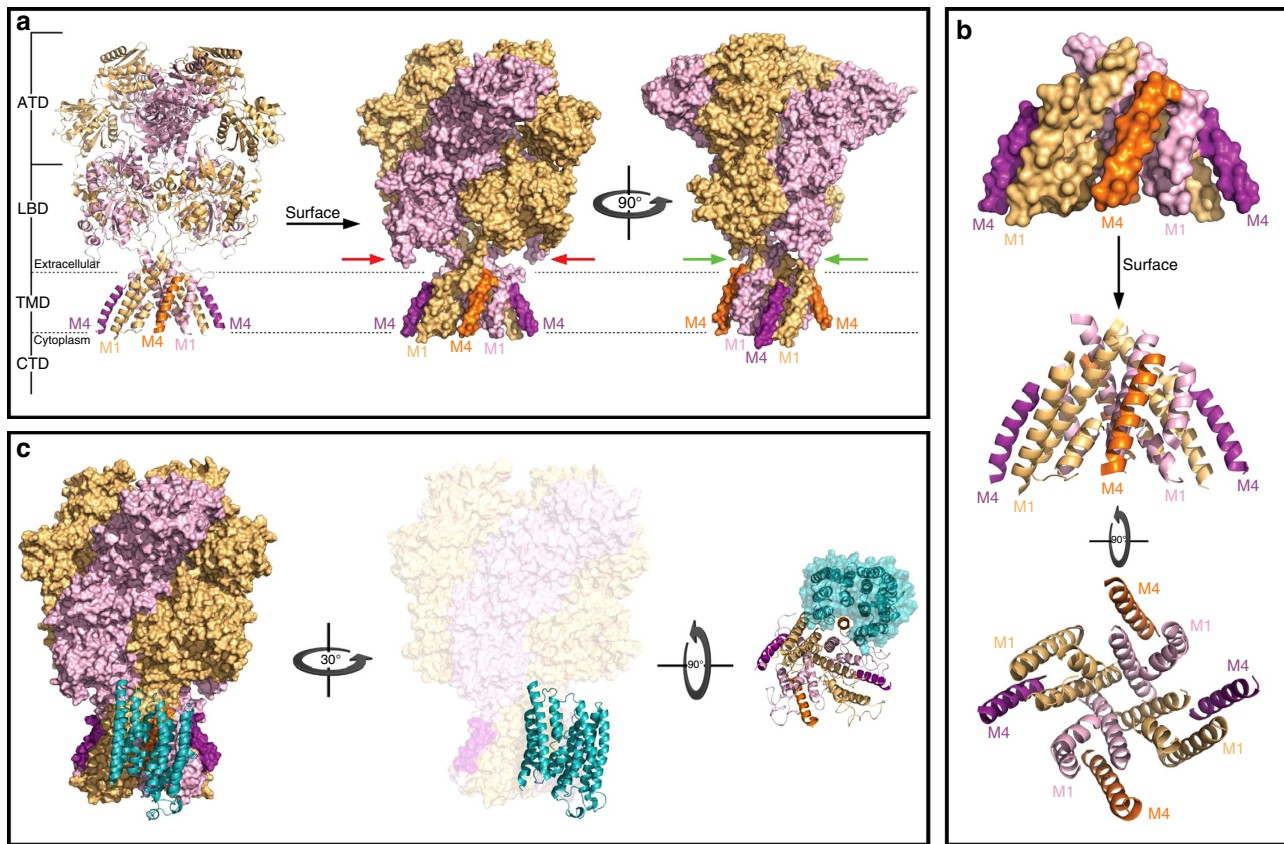

**Figure 2 | Structural basis for docking of ND2 onto the NMDAR TMD. (a)** Overall architecture of the NMDAR. X-ray crystal structure of the NMDAR (PDB code: 4PE5) in cartoon and surface representations with the GluN1 subunit in light orange and the GluN2 subunit in light purple. Note that the intrinsically disordered CTDs are not shown and the protruding M4 of GluN1 and GluN2 are shown in dark orange and dark purple, respectively. Red arrows show the LBD of GluN2 approaching the membrane, preventing the docking of ND2 to the M4 of GluN2. Green arrows show that there is no such occlusion of GluN1–M4 for docking to ND2. **(b)** Organization of the TMD of NMDAR. Lateral and cytoplasmic views of the TMD in cartoon and surface representations, colour-coded as in **a**, revealing how M4 of each subunit lies lateral to the M1 and M3 of the neighbouring subunit. This structural organization results in the protrusion of the M4, and the formation of two shallow grooves between M1 and M4 within one subunit and M4 of one subunit and M1 of the next subunit. **(c)** Cartoon representation of ND2 (cyan) interacting with the TMD of NMDAR (surface representation colour-coded as in **a**) in lateral and cytoplasm view. For the sake of clarity, the cytoplasm view of the complex shows only the TMD of the NMDAR (cartoon) and ND2 surface. Note how M4 of GluN1 has very tight surface complementarity to the TM groove of ND2. See also Supplementary Fig. 1.

localized with GluN1–GluN2A NMDARs (Fig. 3 and Supplementary Fig. 3a). By contrast, ND2 did not co-localize with two other types of co-transfected ligand-gated ion channel; neither GluA1–GluA2 containing glutamate-family AMPA receptors[35], nor non-glutamate family P2X4 receptors[36] showed significant co-localization with ND2 (Supplementary Fig. 3d). ND2 also failed to co-localize with co-transfected PSD95, an NMDAR complex-associated protein (Supplementary Fig. 3c), or with exogenously introduced red fluorescent protein (RFP)-actin (Supplementary Fig. 3d). The PCC values obtained for ND2–GluN1–GluN2A co-localization were significantly different from those of any other proteins tested (Kruskal–Wallis analysis of variance, $P < 0.0001$). Thus, our findings rule out ND2 interacting non-specifically with other ion channels, molecular scaffolds or cytoskeletal elements. Rather, we find there is a specific interaction with NMDARs and that structural features present in NMDARs, but absent in other receptors tested, are necessary for ND2–NMDAR interaction. This specificity of ND2 for NMDARs is consistent with our previous findings[6], and provides validation of this novel co-localization assay.

**ND2 interacts preferentially with GluN1 but not GluN2A.** To test whether the interaction of ND2 with NMDAR requires the presence of both GluN1 and GluN2 subunits, we examined ND2 interaction with each subunit in isolation (Fig. 3b,c). We found that ND2 co-localized with GluN1 when co-expressed alone but not with GluN2A. No significant difference was observed in PCC values between the ND2 and GluN1 or GluN2A versus ND2 and GluN1 alone, suggesting that GluN1 is sufficient for ND2–NMDAR association, confirming our docked model (Fig. 2c).

**GluN1–M4 is required for ND2 interaction.** To determine which regions of GluN1 are required for interacting with ND2, we designed a series of GluN1 deletion mutants (Fig. 4a) and tested their ability to interact with ND2. We targeted domains that contained known protein–protein interaction sites, but whose deletion would not affect subunit trafficking or receptor function. The GluN1 CTD is an intrinsically disordered region known to interact with a wide range of proteins[27,28]. Neither cell surface expression nor basal-level NMDAR function are impacted on CTD removal[9,37]. We therefore generated a GluN1 CTD deletion mutant (GluN1ΔCTD) to assess the contribution of the intracellular CTD to the ND2–GluN1 interaction. We found no significant difference between PCC values for full-length GluN1 and GluN1ΔCTD (Fig. 4b,c). Therefore, GluN1 CTD is not necessary for interaction with ND2.

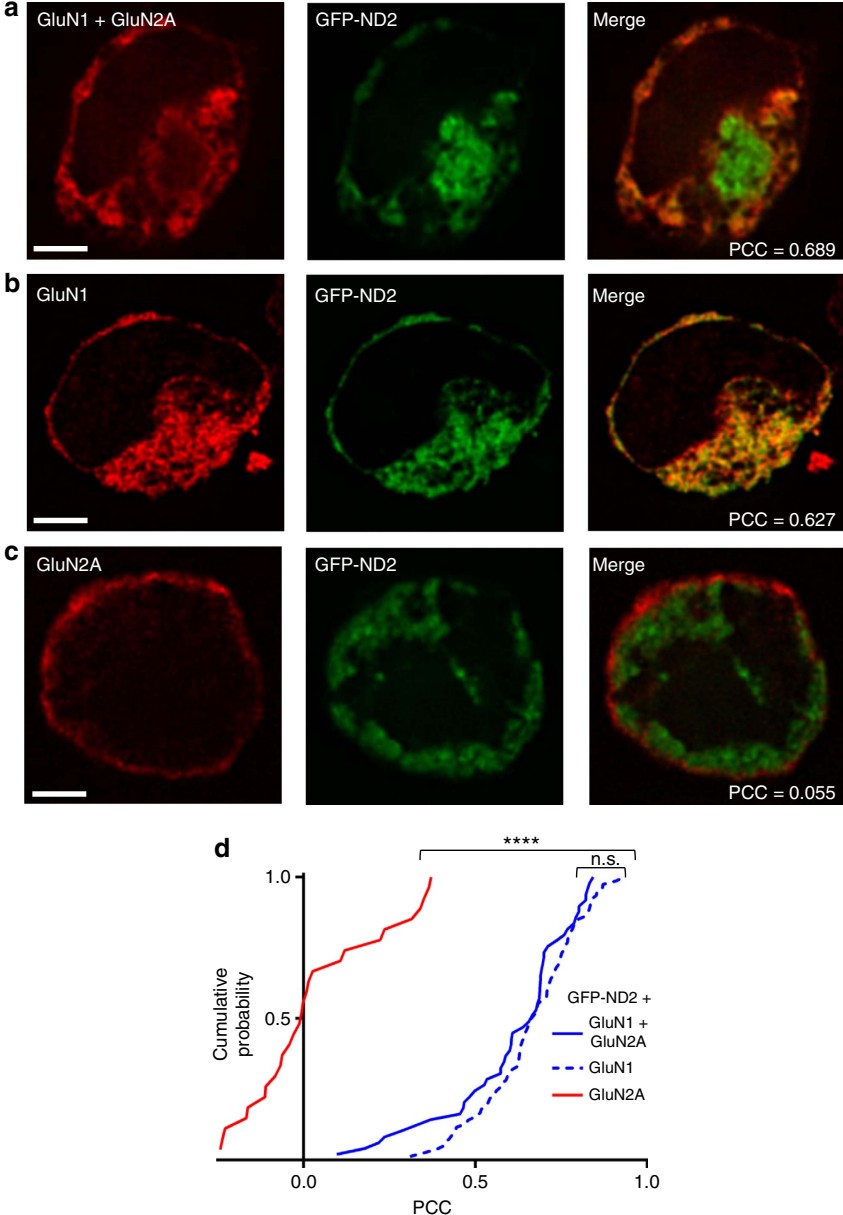

**Figure 3 | GFP-tagged ND2 co-localizes with GluN1 but not GluN2A.** Representative images of HEK293 cells expressing GFP-tagged ND2 with GluN1 + GluN2A (**a**), GluN1 alone (**b**) or GluN2A alone (**c**). (**d**) Cumulative frequency distribution of thresholded PCC values for GFP-ND2 with GluN1 + GluN2A, (mean PCC = 0.61 ± 0.03; $n = 49$), GluN1 alone, (0.67 ± 0.02; $n = 78$) and GluN2A alone (0.03 ± 0.04; $n = 28$). Scale bar, 3 μm. Statistically significant differences between populations are indicated by the symbol '****'($P < 0.0001$), and were evaluated by Kruskal–Wallis non-parametric analysis of variance with Dunn's multiple post hoc comparison tests. $n = $ # of cells. Results are presented as mean ± s.e.m. See also Supplementary Fig. 3.

The ATD is the only other domain of GluN1 known to interact with proteins outside the core subunits[29–31], and therefore we tested whether this domain was required for ND2–GluN1 interaction. As deleting the ATD does not perturb core NMDAR function[38,39] and as we had already determined that the CTD is not required for ND2 interaction, we created an ATD and CTD deletion mutant (GluN1ΔATDΔCTD). We found no difference between PCC values obtained for GluN1ΔATDΔCTD with either full-length GluN1 or GluN1ΔCTD (Fig. 4b,c). Thus, we conclude that neither ATD nor CTD regions of GluN1 are necessary for its interaction with ND2. While GluN2A is necessary for tetrameric assembly of core NMDA receptors, it is not required for ND2 interaction since its addition did not produce significantly different PCC values for either

GluN1ΔCTD or GluN1ΔATDΔCTD compared to that observed for GluN1 truncation constructs alone (Supplementary Fig. 4).

These findings strongly support our model that ND2 interacts with GluN1 TMD, a region not previously known to participate in interactions with proteins other than NMDAR subunits. As suggested by our model, GluN1–M4 is hypothesized to be the primary interacting element. If M4 is necessary then removing it should abrogate the interaction. Therefore, to validate our structural model of the complex, we generated a GluN1ΔATDΔM4ΔCTD construct. We found that removing M4 led to a dramatic decrease in ND2–GluN1 co-localization; there was a significant difference between PCC values obtained for GluN1 subunits lacking M4, GluN1ΔATDΔM4ΔCTD, and those for M4 containing constructs: GluN1, GluN1ΔCTD and

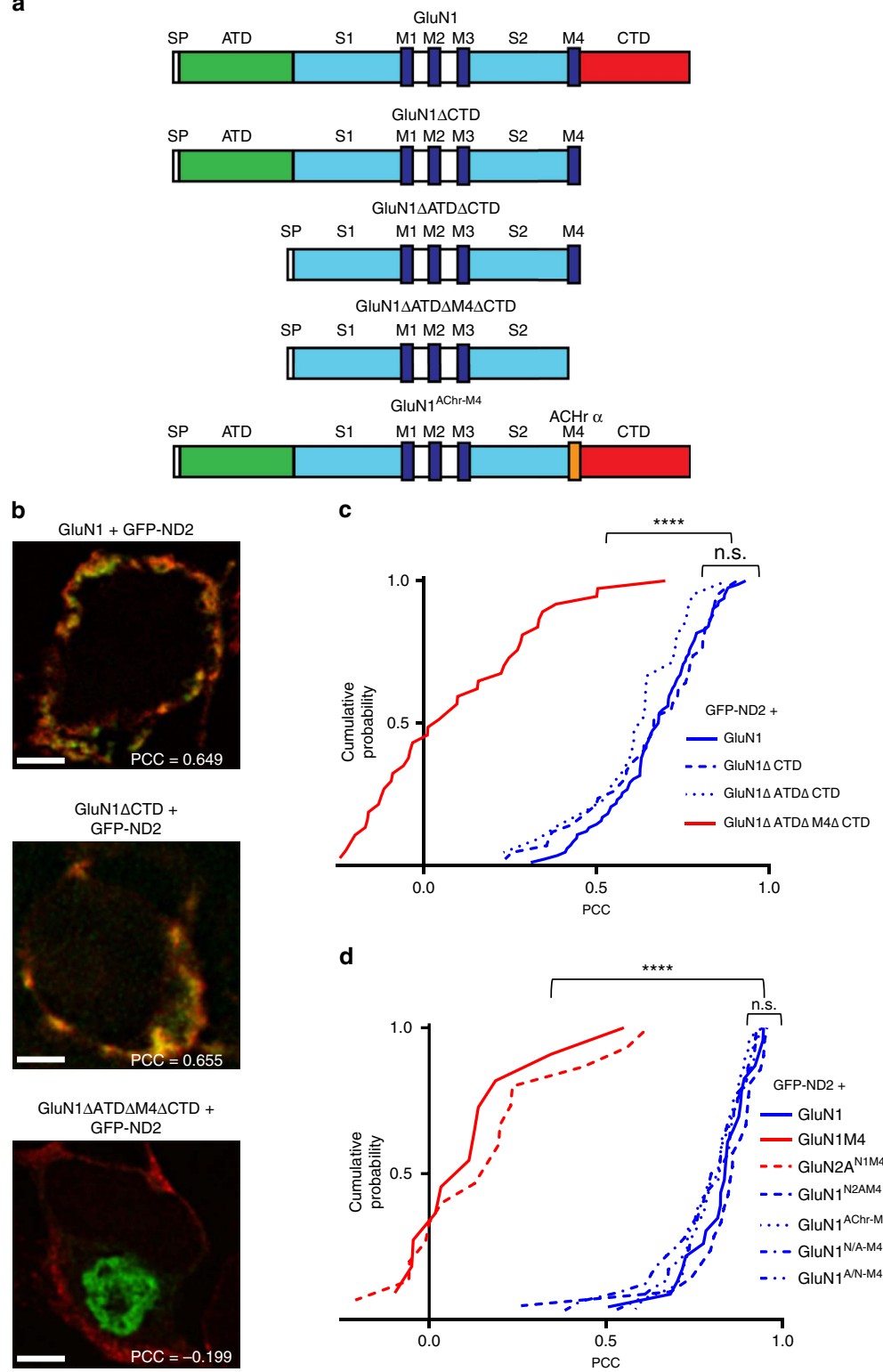

**Figure 4 | GluN1–M4 is necessary for ND2 co-localization.** (**a**) Cartoon representation of GluN1 and subsequent deletion constructs. (**b**) Representative images of HEK293 cells expressing GFP-ND2 + GluN1, GluN1ΔCTD, GluN1ΔATDΔCTD or GluN1ΔATDΔM4ΔCTD. (**c**) Cumulative frequency distribution of thresholded PCC values for GFP-ND2 with GluN1, (mean PCC = 0.67 ± 0.02; $n = 82$), GluN1ΔCTD (0.65 ± 0.03; $n = 42$), GluN1ΔATDΔCTD (0.61 ± 0.03; $n = 21$), and GluN1ΔATDΔM4ΔCTD (0.09 ± 0.04; $n = 37$). (**d**) Cumulative frequency distribution of thresholded PCC values for GFP-ND2 with GluN1, (mean PCC = 0.82 ± 0.02; $n = 23$), GluN1 M4 (0.12 ± 0.06; $n = 11$), GluN2A$^{N1M4}$ (0.17 ± 0.06; $n = 15$), GluN1$^{N2AM4}$ (0.83 ± 0.03; $n = 21$), GluN1$^{AChr\ M4}$ (0.80 ± 0.02; $n = 28$), GluN1$^{N/A\ M4}$ (0.78 ± 0.03; $n = 32$) and GluN1$^{A/N\ M4}$ (0.80 ± 0.02; $n = 29$). Scale bar, 3 μm. Statistically significant differences between populations are indicated by the symbol '****'($P < 0.0001$), and were evaluated by Kruskal–Wallis non-parametric analysis of variance with Dunn's multiple post hoc comparison tests. $n = $ # of cells. Results are presented as mean ± s.e.m. See also Supplementary Figs 4–6 and Supplementary Table 1.

GluN1ΔATDΔCTD (Fig. 4b,c). This loss of co-localization could not be attributed to a lack of expression of either ND2 or GluN1ΔATDΔM4ΔCTD, nor did deletion of M4 appear to drastically alter the distribution of GluN1ΔATDΔM4ΔCTD throughout the cell as compared with other GluN1 constructs (Fig. 4b and Supplementary Table 1). Instead, it appeared that localization of ND2 was altered by the absence of GluN1–M4 as compared with other GluN1 deletion mutants or with the full-length subunit. Consistent with our docking model (Fig. 2c), these results indicate that GluN1–M4 is necessary for interaction with ND2.

To determine whether GluN1–M4, outside the context of other folded GluN1 domains, can interact with ND2, we co-transfected HEK293 cells with GFP-ND2 as well as Myc-tagged GluN1–M4 or GluN2A$^{N1M4}$, in which the portion of native M4 sequence in a full-length GluN2A cDNA was replaced with GluN1–M4 sequences. We found no significant co-localization of GFP-ND2 with GluN2A$^{N1M4}$ (Fig. 4d and Supplementary Fig. 5a) nor with isolated GluN1–M4 (Fig. 4d and Supplementary Fig. 6a). Thus, GluN1–M4 is not sufficient for the ND2 interaction.

We next tested the effect of GluN1–M4 mutations on ND2–GluN1 interactions. GluN1–M4 and GluN2A–M4 share substantial sequence homology (Supplementary Table 1, 31.8% identity, 50.0% similarity, Stretcher software[40]), and the first six amino acids of both M4 regions are identical. In GluN1, a conserved and critical methionine immediately follows these six residues[41], whereas GluN2 subunits have an additional tyrosine before this methionine (Supplementary Table 1). We generated two GluN1–GluN2-M4 chimeras: (i) GluN1$^{insertY818}$ in which a tyrosine was inserted between the highly conserved six-residue segment and the methionine in GluN1–M4 (Supplementary Table 1) with this insertion expected to alter the helical register for all the residues C terminal to this insertion, thereby changing which amino-acid side chains are exposed to the ND2 groove; (ii) GluN1$^{N2AM4}$ in which the entire GluN1–M4 sequence was substituted for GluN2A–M4 to determine the effect of C-terminal residues of GluN1–M4 on ND2 binding. Neither GluN1$^{N2AM4}$ nor GluN1$^{insertY818}$ led to a significant change in interaction with ND2 when compared with native GluN1 (Fig. 4d, Supplementary Figs 5b and 6b).

Another construct, GluN2A$^{ΔY822}$ (Supplementary Table 1), was created to examine whether deletion of the tyrosine immediately before the conserved methionine in GluN2A–M4, resulting in a shift of the M4 helical register to more closely align with that of GluN1, was sufficient to facilitate co-localization of GluN2A with GFP-ND2. The PCC obtained for GluN2A$^{ΔY822}$ was significantly lower than that of GluN1 or GluN1$^{insertY818}$ (Supplementary Fig. 6b), indicating that shifting the helical register of GluN2A–M4 was not sufficient to facilitate ND2–GluN2A interaction. The results obtained for both GluN2A$^{ΔY822}$ and GluN1$^{insertY818}$ M4 helical register mutants suggest that the exact M4 TM region sequence is of secondary importance, and that further point mutations might have minimal effect on ND2–NMDAR interaction.

We then investigated whether a markedly different primary amino-acid sequence at the GluN1–M4 site was sufficient to maintain ND2–GluN1 interaction. For this purpose, we used the acetylcholine receptor α (AChRα) M3 sequence, which is similar in size but has low-sequence similarity to GluN1–M4 (18.2% identity, 36.4% similarity, Stretcher software) and has previously been used in mutagenesis studies to substitute for GluN1–M4 (ref. 42). We generated and tested three GluN1–M4–AChRα–M3 chimeras: (i) GluN1$^{A/N-M4}$ and (ii) GluN1$^{N/A-M4}$, in which C- and N-terminal residues of GluN1–M4 were replaced with C- and N-terminal residues of AChRα–M3, respectively, and (iii) GluN1$^{AChr–M4}$ in which the entire GluN1–M4 was replaced

with AChRα–M3. As shown in Fig. 4 and Supplementary Fig. 5, GFP-ND2 co-localized with all three chimeras. The mean PCC for GluN1$^{AChR–M4}$ was not significantly different from the PCC obtained with another GluN1–M4 substitution mutant GluN1$^{N2AM4}$, or full-length GluN1; GluN1$^{A/N-M4}$ and GluN1$^{N/A-M4}$ also gave PCC values comparable to that obtained for wild-type (WT) GluN1 with GFP-ND2 (Fig. 4). To investigate the function of GluN1 mutant NMDARs, we made whole-cell patch-clamp recordings from HEK293 cells 48 h post transfection with WT or mutant GluN1 constructs together with GluN2A and PSD95. In each case, applying NMDA evoked inward currents that were blocked by D-2-amino-5-phosphonovaleric acid (D-APV), a specific NMDAR antagonist (Supplementary Fig. 7), demonstrating that the GluN1–M4 mutants examined had the capacity to correctly fold into functional tetrameric NMDAR channels that were trafficked to the cell surface. In contrast to previous observations[37], we observed currents with M4-lacking GluN1 receptors (see Supplementary Note 1). Taking these findings collectively, we conclude that the ND2–GluN1 interaction requires the presence of a TM helix at the M4 region, but that variable amino-acid composition is tolerated.

Thus, in the context of the GluN1 subunit, our hypothesis is that the protruding topological surface feature of this fourth TM region, rather than specific primary amino-acid sequence, may be the critical element for GluN1 interaction with ND2, although a generally hydrophobic helical TM segment as found in all these proteins is assumed to be needed. These data are also in agreement with our analysis of the evolutionary conservation of amino acids, which shows that neither the ND2 groove nor the exposed M4 surface of GluN1 are more highly conserved than regions outside the interface (Supplementary Fig. 8).

**TM-6-8 of ND2 is sufficient for GluN1 interaction**. Having identified the critical region in GluN1, we turned to the other side of the ND2–GluN1 interaction and investigated which fragments of ND2 can maintain an interaction with NMDARs. Guided by our ND2 homology model and docked complex structures, we made a series of ND2 deletion mutants that maintained the TM architecture (Fig. 5). Because the helices surrounding the ND2 TM groove (that is, TMs 1, 5, 8 and 11) are not contiguous in amino-acid sequence and with two of these helices being terminal, we could not generate an ND2 construct, other than full length, with an intact groove. Therefore, we divided ND2 into two main fragments based on symmetry of the TM groove: an N-terminal fragment containing TM1-5 and a C-terminal fragment containing TM6-11, both possessing a terminal and a non-terminal groove-forming TM helix. We focused on the C-terminal fragment, which also contains the Src-anchoring cytoplasmic loop, to test whether it was sufficient to direct co-localization with GluN1. All ND2 constructs were co-transfected into HEK293 cells together with GluN1 (Fig. 5b,c). Like full-length ND2, the ND2-TM-6-11 fragment co-localized with GluN1 (Fig. 5b and Supplementary Fig. 9). Dividing ND2-TM-6-11 into two parts, we found the PCC of ND2-TM-6-8 (residues 151–240) (Fig. 5b) is not significantly different from that of TM-6-11, but the PCC obtained for ND2-TM-10-11 (residues 250–347) is significantly reduced (Fig. 5b,c). Furthermore, the ND2-TM-6-8 fragments with or without the cytoplasmic loop connecting TM8 and TM9 (that is, residues 151–240 versus residues 151–223) show similar co-localization (P < 0.05; Fig. 5c). No significant difference in PCC was observed on co-transfection with GluN2A: (i) GluN1 + GluN2A + GFP-ND2-TM-6-8 + loop and (ii) GluN1 + GluN2A + GFP-ND2-TM-6-8 (Supplementary Fig. 10).

Finally, to investigate the role of each TM helix in the interaction with ND2-TM-6-8, we subdivided this region further

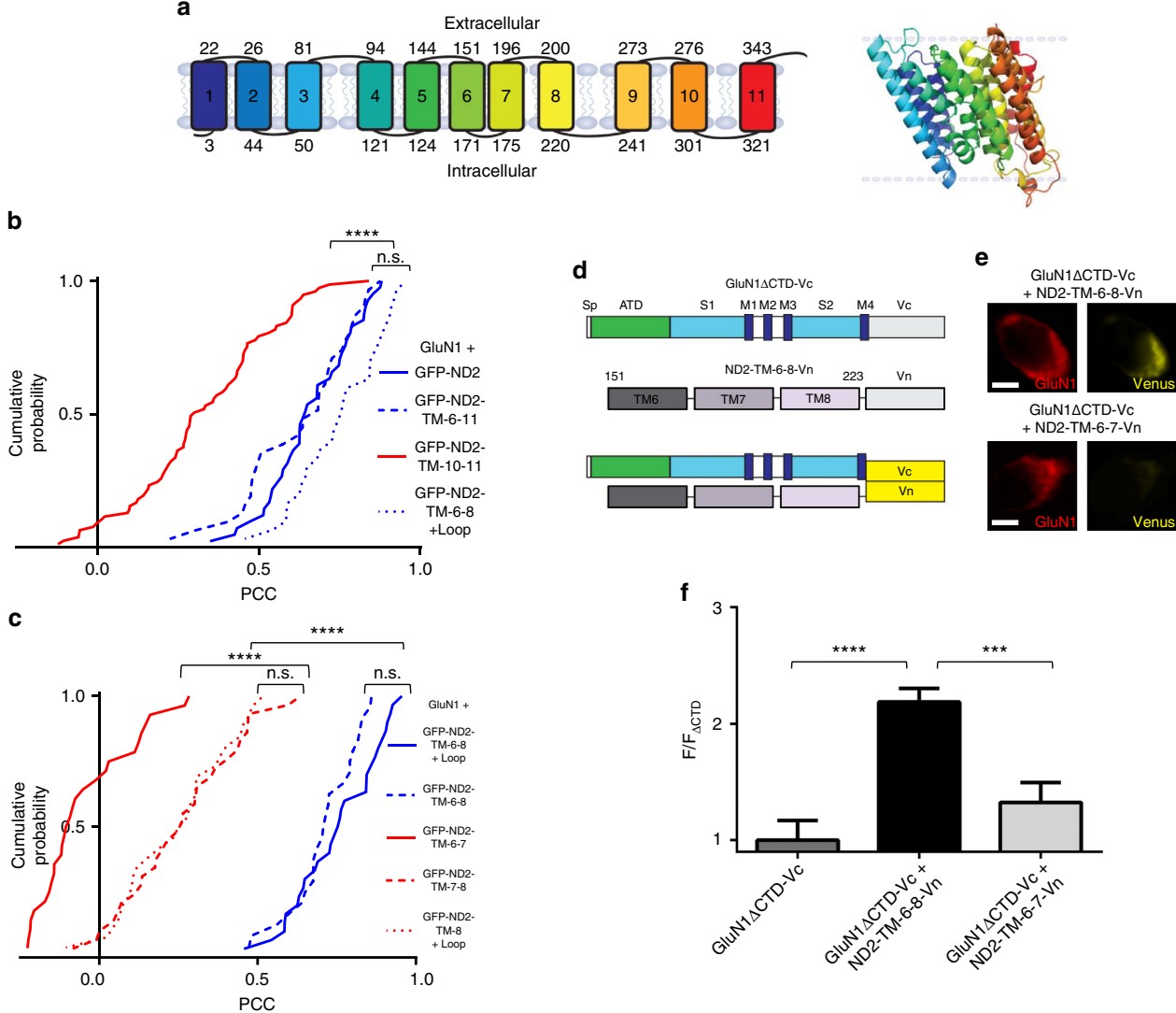

**Figure 5 | ND2-TM-6-8 is the smallest identified region sufficient to co-localize with GluN1.** (**a**) Cartoon model of ND2, with TM regions numbered 1–11. The model was used to rationally design a sequence of GFP-tagged ND2 fragments; ND2-TM-6-11 (residues 151–347), ND2-TM-10-11 (residues 250–347), ND2-TM-6-8 + cytoplasmic loop (residues 151–240), ND2-TM-6-8 (residues 151–223), ND2-TM-6-7 (residues 151–200), ND2-TM-7-8 (residues 175–223) and ND2-TM-8 + cytoplasmic loop (residues 151–240). Inset—homology model of ND2, generated using ND2 homologues from *E. coli* (PDB IDs 3RKO, NuoN subunit) and *T. thermophilus* (PDB code: 4HE8). (**b**) Cumulative frequency distribution of thresholded PCC values for GluN1 with GFP-ND2, (mean PCC = 0.66 ± 0.02; n = 41), GFP-ND2-TM-6-11, (0.63 ± 0.03; n = 31), GFP-ND2-TM-10-11 (0.33 ± 0.02; n = 77) and GFP-ND2-TM-6-8 + cytoplasmic loop (0.75 ± 0.02; n = 30). (**c**) Cumulative frequency distribution of thresholded PCC values for GluN1 with GFP-ND2-TM-6-8 + cytoplasmic loop, (mean PCC = 0.75 ± 0.02; n = 30), GFP-ND2-TM-6-8, (0.71 ± 0.02; n = 24), GFP-ND2-TM-6-7, (0.05 ± 0.03; n = 28), GFP-ND2-TM-7-8, (0.26 ± 0.03; n = 29) and GFP-ND2-TM-8 + cytoplasmic loop (0.25 ± 0.03; n = 27). (**d**) Schematic depicting GluN1ΔCTD-Vc and ND2-TM-6-8-Vn fragments interaction. (**e**) × 100 images of HEK293 cells transfected with GluN1ΔCTD-Vc and either ND2-TM-6-8-Vn or ND2-TM-6-7-Vn constructs. Scale bar, 5 μm. (**f**) Histogram depicting normalized Venus intensity signal observed in GluN1-positive HEK293 cells transfected with GluN1ΔCTD-Vc alone, or GluN1ΔCTD-Vc with either ND2-TM-6-8-Vn or ND2-TM-6-7-Vn. Data were normalized to average Venus fluorescence intensity determined for GluN1-positive HEK293 cells transfected with GluN1ΔCTD-Vc alone (F/F$_{\Delta CTD}$). No significant difference was observed in Venus intensity between GluN1ΔCTD-Vc (1.00 ± 0.17, n = 17) and GluN1ΔCTD-Vc + ND2 ND2-TM-6-7-Vn populations (1.33 ± 0.17, n = 15), but there was a significant difference between GluN1ΔCTD-Vc alone and GluN1ΔCTD-Vc + ND2-TM-6-8-Vn (2.19 ± 0.12, n = 30). Statistically significant differences between populations are indicated by the symbols '***' and '****' (P < 0.001 and P < 0.0001, respectively) and were evaluated by Kruskal–Wallis non-parametric analysis of variance with Dunn's multiple post hoc comparison tests. n = # of cells. Results are presented as mean ± s.e.m. See also Supplementary Figs 9 and 10.

into TM-6-7, TM-7-8 and TM-8 + the cytoplasmic loop and monitored GluN1 co-localization. GFP-ND2-TM-6-8, 6-7 and 7-8 fragments of the predicted molecular mass were detected by immunoblot (Supplementary Fig. 11). Removal of either TM6 (that is, ND2-TM-7-8) or TM8 (that is, ND2-TM-6-7) resulted in a loss of GluN1 co-localization. Additionally, no significant difference in PCC was observed on co-transfection with GluN2A:

(i) GluN1 + GluN2A + GFP-ND2-TM7-8 and (ii) GluN1 + GluN2A + GFP-ND2-TM-6-7 (Supplementary Fig. 10). ND2-TM-8 + the cytoplasmic loop was also insufficient to maintain comparable GluN1 co-localization when compared with TM-6-8 (Fig. 5c). While neither the TM-7-8, TM-10-11 nor TM-8 + loop fragments were found to co-localize with GluN1 to the extent of TM-6-8, they all displayed significantly higher PCC values when

compared with those obtained for ND2-TM-6-7 (Fig. 5b,c). The partial co-localization of this subset with GluN1 likely reflects interaction with the ND2-binding groove TMs, TM8 or TM11. In other experiments we found that GFP-ND2-TM-6-8 co-immunoprecipitated when cell lysates from HEK293 cells transfected with this construct and GluN1 were immunoprecipitated with anti-GluN1 antibody. By contrast, GFP-ND2-TM-6-8 was not immunoprecipitated by nonspecific IgG (Supplementary Fig. 12). Taken together, we conclude that the ND2 region comprising TM-6-8 (that is, amino acids 151–223) is sufficient for significant cellular co-localization with GluN1. These results provide evidence that the groove-forming helices identified in our docked model are required for direct interaction with GluN1 and suggest that helix–helix packing is needed to provide structural and biosynthetic stability[43] to enable GluN1–ND2 binding.

**GluN1–ND2 interaction using BiFC**. As an orthogonal approach to test for an interaction between ND2 and GluN1, we used BiFC[44]. To probe for ND2–GluN1 interaction, we used the yellow fluorescent protein (YFP) variant, Venus, split into N- and C-terminal regions, Vn and Vc, respectively. We generated constructs ND2-Vn and GluN1ΔCTD-Vc by fusing ND2 and GluN1ΔCTD to the N terminus of Vn and Vc, respectively. Similarly, we also generated ND2-TM-6-8-Vn and ND2-TM-6-7-Vn (Fig. 5d) and quantified the fluorescence intensity emitted on reconstitution of Venus in HEK293 cells transfected with GluN1ΔCTD-Vc alone or with ND2-Vn, ND-TM-6-8-Vn or ND2-TM-6-7-Vn (Fig. 5e). We found that cells expressing GluN1ΔCTD-Vc together with ND2-Vn or with ND2-TM-6-8-Vn showed Venus fluorescence (Fig. 5e,f), which was not observed for cells expressing GluN1ΔCTD-Vc or with ND2-TM-6-7-Vn alone. Furthermore, we investigated whether addition of TM-6-8 of ND2 can disrupt the association between ND2-Vn and GluN1ΔCTD-Vc. Therefore, we performed BiFC of ND2-Vn with GluN1ΔCTD-Vc when co-transfected with TM-6-8 of ND2 fused to a non-fluorescing GFP mutant (NF-GFP-ND2-TM-6-8). We observed significantly reduced ND2-Vn-GluN1-Vc fluorescence as compared with co-transfecting with a control pcDNA3 construct (Fig. 6a). Therefore, binding to ND2-TM-6-8 is sufficient to block the interaction of full-length ND2 with GluN1. Taken together, these results confirm the interaction of GluN1 with ND2 or with TM-6-8 but not with TM-6-7 of ND2. This strongly supports both our co-localization results demonstrating interactions between ND2 and GluN1 and the docked model of ND2–NMDAR, highlighting the significant interface with TM8 of ND2.

**ND2-TM-6-8 disrupts Src-mediated current increase**. To test whether ND2-TM-6-8 can interact with native NMDA receptors, we transfected cultured rat hippocampal neurons with GFP-ND2-TM-6-8 (Fig. 6b). After 48 h, we observed in dendrites a distinctive, punctate distribution of GFP-ND2-TM-6-8, which consistently co-localized with GluN1 (Fig. 6b). By contrast, GFP alone or the non-interacting GFP-ND2-TM-6-7 fragment lacked this targeted punctuate distribution pattern, and instead was observed throughout the neurons (Supplementary Fig. 13a–c). These observations suggest that ND2-TM-6-8 is capable of differentially targeting to sites where NMDARs are expressed in hippocampal neurons.

As blocking ND2–Src interaction prevents Src-mediated upregulation of NMDAR function[6], we wondered whether disrupting ND2–GluN1 interaction might similarly suppress Src upregulation of NMDAR current. To test this prediction, we made whole-cell patch-clamp recordings from hippocampal neurons transfected with GFP alone, GFP + GFP-ND2-TM-6-8 or GFP + GFP-ND2-TM-6-7. GFP was added in each case to monitor the successfully transfected neurons due to the relatively poorly fluorescing GFP-ND2 fusion proteins. We induced Src upregulation of NMDAR current by delivering Src-activating peptide, EPQ(pY)EEIPIA, through the recording pipette which induced an increase in NMDAR current in hippocampal neurons, as observed previously[23,45]. We found that EPQ(pY)EEIPIA increased NMDAR current to a similar extent in neurons transfected with either GFP + GFP-ND2-TM-6-7 or GFP alone (Fig. 6c,d). However, the EPQ(pY)EEIPIA peptide failed to increase NMDAR current in neurons transfected with GFP-ND2-TM-6-8; the current instead ran down[19,46] during the recordings (Fig. 6c,d). The blockade of the EPQ(pY)EEIPIA-induced increase in NMDAR current by GFP-ND2-TM-6-8 but not by GFP-ND2-TM-6-7 implies that disrupting ND2–GluN1 interaction prevents Src-mediated upregulation of NMDAR current. Thus, we conclude that GluN1–ND2 interaction is required for Src to upregulate NMDA receptors. Together, our experimental findings using our newly developed co-localization assay and confirming BiFC data provide compelling converging evidence for our docked structural model of an ND2–NMDAR complex defined by TMD-driven interactions between ND2 and GluN1, enabling Src regulation of NMDAR current.

## Discussion

Here we show that GluN1–M4 interacts directly with a groove in ND2. This groove is available in mammalian ND2 due to an evolutionary loss of the first three TM helices of primordial ND2. Furthermore, we determined that ND2-TM-6-8 is sufficient to prevent Src-mediated upregulation of NMDAR currents in neurons. The GluN1–ND2 interaction, mediated by both protein TM regions, is the first such TM–TM interaction described for a core NMDAR subunit and accessory protein, and represents a novel, conserved mechanism for NMDAR activity modulation. We previously characterized the ND2–Src complex by showing that Src N-terminal disordered UD mediates interaction with ND2 residues 239–321 (ref. 6). Our current atomic-level modeling of the ND2–NMDAR complex, along with previous data[6,7], provides the most complete picture to date of the Src–ND2–NMDAR signalling complex (Fig. 7). Our study closes a key gap in knowledge regarding how Src associates with NMDARs to upregulate NMDAR activity. NMDAR upregulation by Src is implicated in a range of physiological and pathological processes. Hence, we anticipate that our present findings defining the critical regions in ND2 and in GluN1 mediating this interaction will be relevant to CNS health and disease.

TM–TM interactions are critical for ion channel pore formation[9,14]. Additionally, TM–TM interactions may regulate function of some ion channels[47,48]. For example, AMPA receptors are dependent on a family of TM AMPA receptor regulatory proteins for plasma membrane trafficking[49–51] and for regulating channel gating[50,52]. By contrast, all previously described interactions involving TMDs of NMDAR subunits have been shown to occur with TMDs of other core NMDAR subunits. No TM–TM interactions have been described with accessory proteins, even if that protein itself contains a TMD[53]. Thus, GluN1–ND2 interaction shares similarity with TM–TM interactions that define the core NMDAR heterotetramer. As such, we suggest that ND2 may be considered an auxiliary NMDAR subunit.

ND2 is synthesized in mitochondria but localizes with NMDARs at PSDs of excitatory synapses[6]. The mechanism by which ND2 is exported from mitochondria has yet to be elucidated but may be through mitochondria-derived vesicles released into the cytosol, which have been shown to shuttle

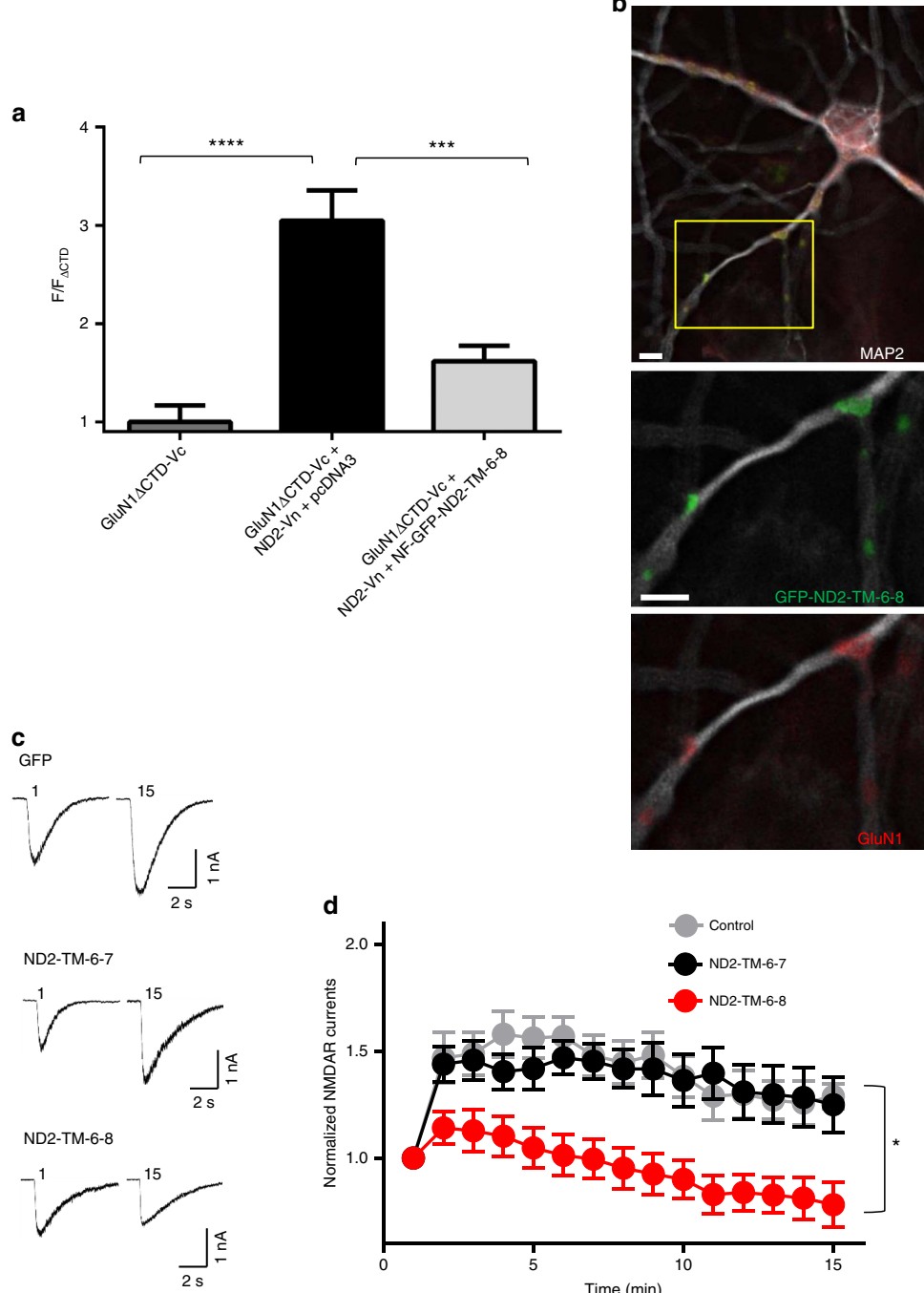

**Figure 6 | ND2-TM-6-8 is sufficient to disrupt the ND2–NMDAR complex.** (**a**) Histogram depicting normalized Venus intensity signal observed in GluN1-positive HEK293 cells transfected with GluN1ΔCTD-Vc and ND2-Vn and either pcDNA3 or the NF-GFP-ND2-TM-6-8 construct. Data were normalized to average Venus fluorescence intensity determined for GluN1-positive HEK293 cells transfected with GluN1ΔCTD-Vc alone $(F/F_{\Delta CTD})$. The observed fluorescence intensity was significantly different between GluN1ΔCTD-Vc + ND2-Vn + pcDNA3 ($3.05 \pm 0.31$, $n = 29$) and both GluN1ΔCTD-Vc + ND2-Vn + NF-GFP-ND2-TM-6-8 ($1.62 \pm 0.16$, $n = 19$), and GluN1ΔCTD-Vc only ($1.00 \pm 0.17$, $n = 17$). Statistically significant differences between populations are indicated by the symbols '***' and '****' ($P < 0.001$ and $P < 0.0001$, respectively), and were evaluated by Welch's one-way ANOVA test. Results are presented as mean ± s.e.m. (**b**) Immunocytochemically stained primary hippocampal neurons transfected with GFP-ND2-TM-6-8. Anti GFP, GluN1 and MAP2 antibodies were used for visualization. Scale bar, 10 μm. (**c**) Representative traces displaying NMDAR currents recorded in primary hippocampal neurons transfected with GFP + GFP-ND2-TM-6-8, GFP + GFP-ND2-TM-6-7 or GFP alone for 48 h. Src-activating peptide (EPQ(pY)EEIPIA) was included in the patch-recording electrode pipette. GFP was added in each case to monitor the successfully transfected neurons due to the relatively poorly fluorescing GFP-ND2 fusion proteins. GFP-ND2-TM-6-7 is known not to interact with GluN1, so was chosen as a control to monitor non-specific effects. (**d**) Grouped data from **c**. ND2-TM-6-8 + GFP ($n = 14$), ND2-TM-6-7 + GFP ($n = 9$) and GFP alone ($n = 6$). Statistically significant differences between populations at 15 min are indicated by the symbol '*' ($P < 0.05$), and were evaluated by Kruskal–Wallis non-parametric analysis of variance with Dunn's multiple post hoc comparison tests. ND2-TM-6-8 + GFP ($n = 9$), ND2-TM-6-7 + GFP ($n = 6$) and GFP alone ($n = 4$) at 15th minute. $n = $ # of cells. Results are presented as mean ± s.e.m. See also Supplementary Fig. 13.

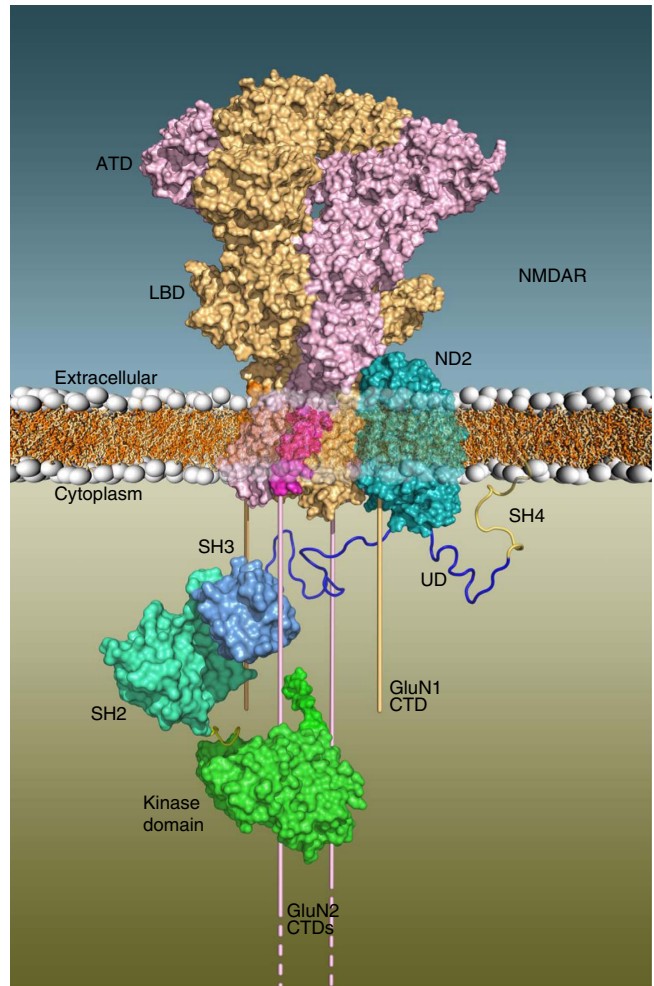

**Figure 7 | Schematic diagram of the ND2–NMDAR complex anchoring Src near the CTDs of GluN2.** Surface representations of the ND2–NMDAR complex in the membrane with the CTDs of GluN1 (light orange) and GluN2 (light purple) shown as bars emerging from the TMD of NMDAR. Src catalytic (green surface), SH2 (light blue surface), SH3 (medium blue surface), UD (dark blue ribbon) and SH4 (gold ribbon) domains are shown, suggesting interactions between the Src UD and cytoplasmic loop of ND2 facilitated by the ND2–NMDAR interaction.

Comparison of evolutionary profiles of ND2 and NMDARs (Supplementary Fig. 2) highlights key events leading to the evolution of ND2-mediated Src upregulation of NMDARs. Higher organisms occupy a clade known as Eumetazoa, which is split into two main groups: bilateria and cnidaria. Bilaterians encode the 'short' form of ND2, and correspondingly have 11 ND2 TM helices. By contrast the 'long' 14 TMD form is found in more primitive organisms, such as prokaryotes, with cnidarians encoding both 'long' and 'short' ND2 (ref. 32). The loss of these three TM helices in higher organisms does not appear to interfere with the electron transport chain function of ND2 and this 'short' form is subsequently conserved in higher organisms[32]. We observed that loss of these helices in the 'short' ND2 homologues results in a TM groove, which serves as the interacting surface for the protruding NMDAR core M4 helix (Fig. 2c). Therefore, emergence of 'short' ND2 proteins likely began before both bilaterian and cnidarian evolution in a common Eumetazoan ancestor, to form a potential interacting groove in ND2. Importantly, in the bacterial complex I the ND2 groove is occluded by its three N-terminal helices, but in the bovine structure the absence of these three helices exposes the ND2 interaction groove laterally[3], and thereby permits binding to GluN1. Furthermore, as NMDAR orthologues have been described in both cnidaria[59,60], and primitive bilateria[59,61], it is believed that NMDARs evolved in a common Eumetazoan ancestor[62]. Both the NMDAR M4 region, and the 'short' binding pocket of ND2, and thus the ND2–NMDAR M4 interaction may have arisen in a common ancestral Eumetazoan line, thereby providing an early mechanism for NMDAR modulation and synaptic plasticity. Loss of the three ND2 TM helices before bilaterian evolution coupled with the NMDAR M4 helix orientation and expansion of the GluN2 CTD leads us to conclude that this evolutionary switch enabled the development of Src-mediated NMDAR modulatory system in higher organisms.

The evolutionary basis for this TMD interaction suggests a shift in our understanding of how mitochondrial and nuclear genomes have evolved, and furthers our appreciation of the complex crosstalk occurring between mitochondrion, cytoplasm and plasma membrane. Given the critical role of NMDARs in CNS health and disease, this new paradigm for receptor regulation may be critical for physiological processes dependant on NMDARs and may facilitate development of novel therapeutics for CNS disorders.

## Methods

**Modelling of the ND2–NMDAR complex.** A homology model of ND2 was obtained using the Phyre2 server with default parameters[63]; this server facilitates modelling using the Modeller software[64], including searching for homologues, sequence alignment and building the model. The resulting model was primarily based on the crystal structures of the membrane domain of respiratory complex I from *E. coli* (3.0 Å resolution) and *T. thermophilus* (3.3 Å resolution) (PDB IDs 3RKO (chains L, N and M) and 4HE8 (chains T, I and M), respectively); the sequence alignments used with the sequence similarities (ranging from 34 to 45%) and identities are shown in Supplementary Fig. 14 and Supplementary Table 2. Phyre2 reported 100% confidence in the homologous nature of the protein sequences used for modelling, a good indicator of a strong model. The crystal structure of GluN1A–GluN2B NMDAR (PDB ID: 4PE5) (ref. 15) was used for docking with ND2 after filling in the missing residues lacking electron density using a homemade crystallography and NMR systems - CNS script[65]; note none of these added residues were found to interact in the docked model. The ND2–NMDAR docking process was guided by the assessment of the surface complementarity between the TMD of NMDAR and the ND2 homology model, assessed using molecular visualization with PyMol (The PyMOL Molecular Graphics System, Version 1.7.4 Schrödinger, LLC), focusing on steric and electrostatic properties. Steric/topological complementarity was observed only for the protruding GluN–M4 helix and the ND2 groove with no obvious electrostatic complementarity within the primarily hydrophobic TM regions. We generated in-house docking scripts to perform the ND2–NMDAR docking using the following procedures: First, the third missing N-terminal TM helix of NuoN

mitochondrial proteins to both peroxisomes[54] and lysosomes[55]. It is possible that ND2 is first exported from mitochondria and subsequently interacts with GluN1 in the endoplasmic reticulum, where GluN subunits are assembled, or in the Golgi and the ND2–NMDAR complex is then transported to synapses. Indeed our experiments, using nuclear codons for both ND2 and GluN constructs and hence translated extramitochondrially, demonstrate that ND2 and GluN1 may assemble outside the mitochondria. It is nevertheless possible that Src–ND2–NMDAR ternary complex assembly may alternatively occur within the mitochondria itself. NMDA receptors have been found in neuronal mitochondria[56] and Src has also been shown to localize to mitochondria[57], including at respiratory complex I (ref. 58). This presents an intriguing possibility that constituents of the Src–ND2–NMDAR complex may interact in neuronal mitochondria and then exported to the cell membrane. Additionally, due to the ubiquity of mitochondria and thereby ND2, we suggest that ND2 may be present not only at synapses, but in all heterologous expression systems used to assess NMDAR activity.

(whose absence in human ND2 results in the ND2 groove) was aligned with M4 of GluN1 and NuoN was rotated about TM4 to minimize steric clashes. Next, ND2 was positioned by aligning it on NuoN. Finally, all side chains were removed from ND2 and NMDAR before putting them back using a crystallography and NMR systems - CNS script that performs short dynamics and minimizations for reorienting the side chains and removing steric clashes. The contact surface area between ND2 and the NMDAR was obtained by taking the difference between the accessible surface area in the combined isolated proteins (the ND2 model plus the NMDAR structure) and the docked complex using the get_area command in Pymol. Our docking scripts along with the coordinates of the ND2 homology model and the ND2–NMDAR complex are available on request.

Analysis of the evolutionary conservation of the amino acids in ND2 and NMDAR was performed on the ConSurf server (http://consurf.tau.ac.il/)[66] using 95% maximal identity between sequences and 35% minimal identity for homologues (Supplementary Fig. 8).

**Cell culture and transfection.** HEK293 cells ($3 \times 10^4$ cells per cm$^2$) were plated onto 12-well culture dishes containing poly D-lysine coated glass coverslips for the co-localization experiments, and 35 mm culture dishes for the electrophysiology experiments. HEK293 cells (ATCC, Virginia, USA) were cultured with Dulbecco's modified eagles media (Invitrogen, Burlington, ON, Canada) supplemented with 10% foetal bovine serum (Invitrogen) and 1% penicillin–streptomycin (Wisent, St Bruno, QC); 37 °C, 5% CO$_2$ FuGene HD (Promega BioSciences, LLC. Sand Luis Obispo, CA, USA) was used for transfections, including GFP-ND2 constructs, at a ratio of 1:12 (µg DNA:µl reagent). Turbofect (Thermo Fisher Scientific) was used for the electrophysiology transfections, at a ratio of 1:1 (µg DNA:µl reagent), and GFP was co-transfected to allow identification of positively transfected cells. For experiments shown in Supplementary Fig. 7, PSD95 was co-transfected to increase the likelihood of observing NMDAR currents for the GluN1–M4 mutants, as PSD95 has been shown to increase NMDAR currents in heterologous systems[67]. After transfection, cells were maintained in Dulbecco's modified eagles media supplemented with 10% foetal bovine serum for 48 h before immunocytochemical experiments. D-APV (500 µM; Tocris, Minneapolis, MN, USA) was included in all transfections containing GluN1 and GluN2A subunits.

**Molecular biology.** Mammalian expression vectors encoding WT rat GluN1-1a, GluN2A cDNAs were gifts from J. MacDonald (University of Toronto, Toronto, ON, Canada) and rat PSD95 cDNA was generously provided by M. Sheng (MIT, Cambridge, MA, USA). The construction of a non-mitochondrially encoded variant of human ND2 was generated within the Salter lab[6], and was subcloned into an enhanced GFP expression vector, pEGFP-C1 (Clontech, Mountain View, CA, USA), to generate the N-terminally tagged fusion construct GFP-ND2. GluN1, GluN2A and GFP-ND2 mutants were generated using the Change-IT site-directed mutagenesis kit (Affymetrix, Santa Clara, CA, USA) (Supplementary Table 3). After mutagenesis, the restriction enzymes PvuI (NEB #R0150S) and BamH1 (NEB #R0136S) were used to engineer the GluN1ΔATDΔCTD and GFP-ND2-TM-7-8 constructs, respectively. The non-fluorescent GFP-ND2 mutant NF-GFP-ND2-TM-6-8 was created by deletion of amino acids 1–7 of the GFP protein in the GFP-ND2-TM-6-8 construct, as per strategy employed by Li et al.[68] All ATD deletion constructs were created by deletion of the ATD (amino acids 19–345) after the GluN1 signal peptide coding region (amino acids 1–18). The ΔCTD and ΔM4ΔCTD deletion constructs were created by insertion of a stop codon at position 837 or 813, respectively.

The GluA1 and GluA2 constructs were a generous gift from Dr Zhengping Jia, and the BiFC leucine zipper constructs (Zipper-Vn and Zipper-Vc) a gift from Dr Christophe Altier. RFP-tagged Actin (LifeAct) was obtained from Ibidi (Munich, Germany), and the P2X4R construct from ATCC (Manassas, Virginia). The GFP-ND2-TM-6-11, GFP-ND2-TM-10-11, GFP-ND2-TM-6-8 + loop, GluN1–M4, GluN2A–M4, GluN1ΔCTD-Vc (GluN1ΔCTD fused to the C-terminal portion of yellow fluorescent protein Venus), ND2-Vn, ND2-TM-6-7-Vn, ND2-TM-6-8-Vn (full-length ND2, ND2 (151–200) or ND2 (151–223) fused to the N-terminal portion of yellow fluorescent protein Venus) constructs were generated by Genscript, NJ, USA. All constructs were verified by DNA sequencing.

**Primary cell culture.** Hippocampus cultures were prepared from timed pregnant Wistar rats. E16-E17 foetuses were decapitated and transferred to chilled Hank's solution. A T-shaped incision was made in each head along the sagittal suture and inter-aurally, the calvaria was flipped back and the exposed brain was lifted out. The hemispheres were dissected apart and the cerebellum was discarded. The meninges were peeled from each hemisphere. The hemisphere was laid medial side up and the hippocampus was dissected out. The hippocampi were pooled and the tissue was mechanically dissociated through a 100 µl pipette tip. The cells were plated onto poly-D-lysine-coated glass coverslips. The culture media was Neurobasal medium supplemented with foetal bovine serum, L-glutamine and B-27 supplement. These procedures have been approved by the Animal Care Committee at The Hospital for Sick Children (SickKids).

**Immunocytochemistry.** Transfected HEK293 cells and cultured hippocampal neurons grown on coverslips were washed in phosphate-buffered saline (PBS) and

fixed using 4% paraformaldehyde in PBS for 15 min. Cells were washed in PBS, and then permeabilized by incubation in 0.1% Triton for 5 min. Subsequently, the cells were washed in PBS and blocked in 10% normal donkey serum for 1 h. The coverslips were then incubated with combinations of the following antibodies in 1% normal donkey serum: anti-mouse GluN1 (1:1,000, BD Biosciences, Cat. No. 556308); anti-mouse GluN2A, (1:1,000, BD Biosciences, Cat. No. 612286); anti-rabbit GluA1 (1:500, Calbiochem, Cat. No. PC246); anti-rabbit P2X4R (1:500, Alomone Labs, Cat. No. APR-002); anti-mouse PSD95 (1:500, BD Biosciences, Cat. No. 610496); anti-chicken MAP2 (1:4,000, Covance, Cat. No. pck554p). All coverslips were incubated at 4 °C overnight, washed in PBS (3 × 10 min) and incubated in fluorescently conjugated secondary antibodies in 1% normal donkey serum as appropriate; Cy3 donkey anti-mouse (1:4,000, Jackson, Cat. No. 715-165-150), Cy3 donkey anti-rabbit (1:4,000, Jackson, Cat. No. 711-165-152), Cy5 donkey anti-chicken (1:4,000, Jackson, Cat. No. 703-175-155) for 2 h at room temperature. Hundred micromolar Hoescht was added for the final 15 min of incubation to stain for nuclei. Following a final wash in PBS (3 × 10 min), the coverslips were mounted on saline-coated slides (Sigma) with Fluoromount (Sigma). GFP fluorescence was consistently observed in HEKs transfected with full-length GFP-ND2 or GFP-ND2 fragments, and only fluorescing cells were selected for analysis.

**Widefield microscopy.** Widefield HEK293 images were collected using a × 100 1.45NA objective, by a Photometrics QuantEM 512SC or Hamamatsu Flash 4.0 camera with a Zeiss Axiovert 200M microscope (Carl Zeiss Microscopy, Germany) using the Volocity software (version 6.0; Perkin Elmer). Final processing was performed on deconvolved images with Adobe Photoshop CS5 without changing the original resolution and colour depth. Thresholded PCC values as described by Barlow et al. (2010) were obtained using the Volocity co-localization function. All imaging was carried out blinded.

**Electrophysiology.** Cultured 12–15 days old hippocampal neurons were transfected with GFP alone, GFP + GFP-ND2-TM-6-7, or GFP + GFP-ND2-TM-6-8 for 48 h, and subsequently whole-cell patch-clamp recordings were undertaken at room temperature. The extracellular solution consisted of (in mM, pH 7.35): 140 NaCl, 5.4 KCl, 25 HEPES, 25 glucose, 1.3 CaCl$_2$, 0.001 glycine, and 0.0005 TTX. Recording electrodes (4–7 MΩ) were pulled from thin-walled glass (World Precision Instruments) using a P-87 pipette puller (Sutter Instrument Company) and filled with the intracellular solution composed of (in mM, pH 7.25): 130 CsF, 10 CsCl, 10 BAPTA, 10 HEPES, 4 Mg-ATP. NMDAR-mediated currents were evoked by puff application of L-aspartic acid sodium salt (dissolved in the extracellular solution, Sigma) at 250 µM for 50 ms using a Picospritzer II (General Valve Corporation). Whole-cell patch-clamp recordings were also undertaken at room temperature on HEK293 cells transfected with GluN2A, PSD95 and WT or mutant GluN1. NMDAR-mediated currents were evoked by NMDA (Sigma, USA) at 250 µM for 2 s using fast-step perfusion system (SF-77B, Warner Instruments; glycine at 10 µM was included in the recording solution under this condition). Currents were recorded only from GFP fluorescent cells at the holding potentials of − 60 mV using an Axopatch-1D amplifier (Molecular Devices) for both hippocampal neurons and HEK293 cells. The electrical signals were filtered at 2 Hz, and the recording data analysed off-line using Clampfit software (Molecular Devices). NMDAR-mediated currents were normalized to the first response (100%). EPQ(pY)EEIPIA peptide (2 mM; GenScript Company) was dissolved fresh immediately before the experiments.

**BiFC assay.** HEK293 cells were transfected with GluN1ΔCTD-Vc alone or with ND2-Vn, ND2-TM-6-8-Vn or ND2-TM-6-7-Vn. The non-fluorescent GFP-ND2-TM-6-8 fusion (NF-GFP-ND2-TM-6-8) or pcDNA3 constructs were used to test the disruption of the ND2-Vn and GluN1ΔCTD-Vc interaction. To optimize the generation of the Venus signal, each of the Vc and Vn fragments was fused to a Leucine zipper motif. This sequence is known to dimerize[69] and thereby lead to a strong association between Vn and Vc fragments, resulting in Venus fluorescence. HEK cells were transfected with Zipper-Vn and Zipper-Vc to optimize the maximum for Venus signal measurement. HEK cells were transfected with GluN1ΔCTD-Vc and Zipper-Vn to determine background level of cell fluorescence. Cells were fixed, permeabilized and Cy3 labelled with anti-mouse GluN1 antibody (as above). Subsequently, images were collected using a × 20 objective, by a Photometrics QImaging Retiga 2000r camera with a Leica DM2500 fluorescent microscope (Leica Microsystems, Germany) using the Image Pro Premier software (version 9.0; Media Cybernetics). Cy3-positive (GluN1ΔCTD-Vc -containing) cells were identified (3.5 × cell background intensity), and the presence or absence of a Venus signal (> 3.5 × or < 3.5 × cell background intensity, respectively) assessed by Photoshop CS5 software (version 12.0.4, Adobe Systems Incorporated). All values were normalized to GluN1ΔCTD-Vc fluorescence. All imaging was carried out blinded.

**Data analysis.** Data were analysed for significance using Prism (Graphpad Software) with the Kruskal–Wallis non-parametric analysis of variance and Dunn's multiple post hoc comparison tests utilized for the cumulative frequency distribution, electrophysiological and BiFC data in Fig. 5f. Welch's one-way

ANOVA test (Microsoft Excel, 2010) was used for the BiFC data in Fig. 6a. Statistical significance was established at $P < 0.05$, '*'. Results are presented as mean ± s.e.m. Pairwise alignment of GluN1–M4 and AChRα M3 domains was undertaken using Stretcher software, http://www.ebi.ac.uk/Tools/psa/emboss_stretcher/[40].

**Western blotting.** Transfected HEK293 cells were lysed by modified RIPA buffer (50 mM Tris, pH 8.0, 150 mM sodium chloride, 1% NP-40, 0.5% sodium deoxycholate, 0.1% SDS and 2 mM EDTA) supplemented with complete protease inhibitor cocktail (Roche). Protein content was determined by BCA protein assay (Thermo Fisher). Samples (10–30 ng protein) were separated by SDS–PAGE using a 4–15% gradient gel (Bio-Rad) and transferred to a polyvinylidene difluoride membrane. The membrane was blocked for 1 h in Odyssey Blocking buffer (Li-Cor) at room temperature and incubated overnight at 4 °C with anti-GFP antibody (1:250) diluted in 1:1 TBS:Blocking buffer. Incubation in secondary antibody, (1:5,000), was performed the next day for 90 min at room temperature in 1:1 TBS:Blocking buffer solution. Primary antibody used was rabbit anti-GFP (Genetex, GTX20290). Secondary antibody used was goat anti-rabbit IRDye 800CW, (Li-Cor, #925-32211). Proteins were visualized by an Odyssey Imaging system, model 9120 (Li-Cor).

**Co-immunoprecipitation.** HEK293 cells (3 × 35 mm dishes) were transfected with GFP-ND2 TM 6–8 and GluN1 constructs, and 48 h later lysed with 200 μl modified RIPA buffer per dish. The lysate was centrifuged for 10 min at 14,000 g, the supernatants combined, and 250 μl supernatant + 750 μl RIPA buffer incubated with 2 μg of either mouse anti-GluN1 (BD Biosciences, Cat. No. 556308) or nonspecific mouse IgG (Sigma, Cat. No. 12–371) overnight at 4 °C. Hundred microlitre of protein G-Sepharose beads (GE Healthcare) (50% slurry) was added to each, and the samples incubated on a rotator for 4 h at 4 °C. Immunoprecipitates were then washed three times with RIPA buffer, resuspended in 40 μl SDS–PAGE loading buffer, and boiled for 5 min. Samples (20 μl lysate per immunoprecipitate) were separated by SDS–PAGE, and western blotted as described above. Primary antibody used was rabbit anti-GFP (Genetex, GTX20290). Secondary antibody used was goat anti-rabbit IRDye 680, (Li-Cor, #926-32221).

**Code availability.** The custom crystallography and NMR systems - CNS scripts used in the process of docking the ND2 onto the NMDAR crystal structure, as well as the homology model of human ND2 and the docked model between ND2 and NMDAR are also available from the authors on request.

**Data availability.** Data supporting the findings of this study are available within the article and its Supplementary Information files and from the corresponding authors on reasonable request. The following PDB IDs were used in this work: 3RKO, membrane domain of complex I from *E. coli*; 4HE8 and 4HEA, membrane domain of complex I from *T. thermophilus*; 4PE5, GluN1–GluN2B NMDA receptor ion channel. The following NCBI GenBank accession numbers were used in this work: *Grin1* (GluN1), NM_001270602.1; *Grin2A* (GluN2A), NM_012573.3; *Dlg4* (PSD95), NM_019621.1; *Gria1* (GluA1), NM_031608.1; *Gria2* (GluA2), NM_017261.2; *P2RX4* (P2X4R), NP_002551.2. The following UniProtKB accession number was used for *MT-ND2* (ND2), P03891.

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

## Acknowledgements

We thank Zheng Ping Jia for the GluA1 and GluA2 constructs, and Christophe Altier for providing the BiFC leucine zipper vectors. We also thank Gregory Borschel for the use of the Leica DM2500 fluorescent microscope. We greatly appreciate the technical assistance provided by Ameet Sengar, Lu Han, Simon Beggs, Erika Harding, Graham Pitcher, Vivian Wang, David Wong, Michael Woodside, Paul Paroutis, Mike Willand, Vishaal Rajani and Janice Hicks. We thank Ameet Sengar for comments on the manuscript. This study was supported by grants from CIHR to M.W.S. (MT-12682) and J.D.F.-K. (MOP 114985) and from the Ontario Research Fund research excellence programme to M.W.S. and J.D.F.-K. and a CIHR post-doctoral fellowship to A.B. M.W.S. holds a tier 1 Canada research chair in neuroplasticity and pain and J.D.F.-K. holds a tier 1 Canada research chair in intrinsically disordered proteins. M.W.S. holds the Northbridge Chair in Paediatric Research.

## Author contributions

D.P.S., A.B., M.K., W.Z. and H.L.L.-P. performed the experiments and modelling for this work. Y.N.D. created the GFP-ND2 construct. D.P.S., A.B., M.K., J.D.F.-K. and M.W.S. designed the research. D.P.S., A.B., M.K., W.Z., J.D.F.-K. and M.W.S. wrote the manuscript. J.D.F.-K. and M.W.S. supervised the project.

## Additional information

**Competing interests:** The authors declare no competing financial interests.

