## [Peer Review File · Nature Communications]

Reviewers' Comments:

Reviewer #1 (Remarks to the Author):

The manuscript presents compelling evidence that the interaction between ND2 and the NMDAR complex is mediated through the transmembrane domain. A molecular model of the complex is derived by (a) homology modeling of ND2 followed by (b) docking of the M4 transmembrane helix of NMDR to the ND2 homology model. The model guides experimental tests of the interaction using mostly colocalization assays. The examination includes both intact proteins and deletion mutants, missing various subunits and domains. The simplest interpretation of the results is that indeed the interaction is mediated through the transmembrane region. Tracing the evolutionary profile of ND2 shows that metazoans feature 'long' variant that includes 3 additional transmembrane helices that are missing in higher organisms. These helices block the M4 docking site, suggesting that the interactions, which ultimately allows upregulation of the complex, emerged late in evolution. This is a very interesting and carefully conducted study that significantly advances the field. That computations guided experiments is particularly appealing to me. The comments listed below may further improve the quality of the work and presentation.

Specific comments:

- 1) Analysis of the evolutionary conservation of the amino acids on both sides of the transmembrane interface, for example using ConSurf (<http://consurf.tau.ac.il/>), may further validate the molecular model. The anticipation is that the interface would be more highly conserved than the periphery. This approach has been used, for example, by Gofman et al 2012, Structure DOI: 10.1016/j.str.2012.05.016 (for examination of a different docking complex).
- 2) How about testing the effect of point mutations in key residues on the interaction? It would provide validation of the molecular details of the docking model, which is tentative otherwise. For example, the rotation angle of the M4 helix around its axis might be wrong. The ConSurf analysis above may highlight specific amino acids to mutate.
- 3) The results seem to suggest that the exact nature of the transmembrane region is of secondary importance, which is surprising in view of the specificity observed. If so, the anticipation is that the interface might be less evolutionarily conserved, and point mutations might have little or no effect on the interaction. It is worthwhile testing.
- 4) Regarding the evolutionary scenario, do higher organisms carry also the long NMDAR variant, or was it completely lost in evolution?
- 5) The modeling procedure used makes sense, but alternatives should be examined as well. In particular:
 - (a) Other tools for modeling the ND2 structure should be used. Two options that are easily accessible are HHpred (<http://toolkit.tuebingen.mpg.de/hhpred>) and FFAS03 (<http://ffas.sanfordburnham.org/ffas-cgi/cgi/ffas.pl>).
 - (b) Similarly, also more tools should be used for docking. For example, PatchDock (<http://bioinfo3d.cs.tau.ac.il/PatchDock/>) and GRAMMX (<http://vakser.compbio.ku.edu/resources/gramm/grammx/>).
- 6) After validation, the coordinates of the model structure should be made publically available to examine further tests, and usage, by others.
- 7) There are many abbreviations, which complicated my reading of the paper. Maybe some could be omitted. For example, PSD?
- 8) I would add a figure of the whole complex in the main text to make it easier for the reader to follow the various interactions. Some is presented in Fig. 7, but it comes too late to my taste.

Reviewer #2 (Remarks to the Author):

The manuscript by Scanlon et al describes direct interaction of NADH dehydrogenase (ND2) with

NMDA receptors. The paper elaborates the previous finding that ND2 facilitates upregulation of NMDA receptor activity by Src. The work takes a rather bold approach to predict the mode of protein-protein interaction by molecular model docking. The authors support their interaction model by observing protein co-localization using fluorescence microscopy. By using truncated version of NMDA receptor subunit, GluN1, and the truncated ND2 constructs, the authors delineated the interaction site to be TM4 of GluN1 and TM6-8 (likely TM8) of ND2. They further support this observation by showing the attenuation of Src-mediated upregulation of NMDA receptor current by electrophysiology. The mechanism of NMDAR regulation by its cytoplasmic domain, for example by Src, has been extensively pursued by the field, but no convincing result has been published, thus far. The biology of NMDAR regulation, which is heavily implicated in signal transduction of neuronal plasticity as well as neuronal diseases, is undoubtedly an important subject. This work also reports the first potentially transmembrane interacting proteins with NMDA receptors somewhat reminiscent of TARP proteins for AMPA receptor family. While the crude microscopic method to speculate on protein-protein interaction is a good first step, the work needs to be further supported by biochemistry such as pull-down experiments to directly probe protein-protein interaction.

-Fluorescent microscopic method to probe localization of ND2 and NMDARs is a good start. This work requires some biochemical experiment (IgG-based pull-down assay) to confirm that direct interaction occurs. There has to be some correlation is with the crude microscopic observation and the actual protein-protein interaction. If ND2 and NMDARs are interacting as the authors suggested through modeling, ND2 should be pulled down convincingly by anti-GluN1 and anti-GluN2 antibodies. Alternatively, interaction of ND2 with purified NMDAR proteins may be a very convincing option.

-Regarding data in Fig. 4 and 5, truncation of the transmembrane domains in GluN1 and ND2 is extreme. Do authors know that truncation of TMD4 in GluN1 expressed alone is properly folded? Or, in the case of ND2, do authors know that the proteins are even expressed? Could the result that TMD4-truncated GluN1 does not interact with ND2 caused by non-tetrameric assembly of this construct? In all of the experiments, the authors should co-express GluN2 subunit along with the GluN1 mutants since expression of GluN1 alone does not represent a physiological state.

-As an alternative approach to truncation, can authors mutate GluN1-TMD4 to alter interaction with ND2? The authors may test the mutant constructs for the NMDAR activity and assess interaction with microscopy and pull-down.

-Fig.5a, More detail on membrane topology would help - the authors should show which side of the schematics is extracellular and cytoplasmic.

-Fig.6C. It appears that the decay time of aspartic acid induced current is significantly slower in ND2 TM6-8 compared to ND2 TM6-7 or GFP control. Could it possibly be the difference between ND2 associated NMDARs and non-associated NMDARs? Would the authors see the difference in decay time without Src activating peptides?

-Related to the above question, it would be worth conducting an experiment where NMDAR (for example, GluN1/GluN2A or B) is co-expressed with ND2 in HEK cells and deactivation and desensitization time course is measured. There is a possibility that ND2 may act like the NMDAR-equivalent of TARP and Neto2 for AMPA and kainate receptors.

Reviewer #3 (Remarks to the Author):

Scanlon et al. is a highly novel and quite interesting study that evaluates structural basis of the interaction between ND2 and NMDARs, and the impact of this interaction for Src-mediated upregulation of NMDA receptor currents in neurons. This study is the first convincing evaluation of a potential auxiliary NMDAR protein with significant interaction mediated by the core transmembrane domain of the receptor, which makes the work significant and timely. The

experimental designs are clever and well-executed, and the study makes excellent use of modelling, molecular biological approaches, and imaging. The authors nicely demonstrate how ND2 interacts with GluN1 but not GluN2 due to steric hindrance of the LBD, and go on to demonstrate functionally that the interaction is relevant for Src signalling in hippocampal neurons. The manuscript is well written, easy to understand, and thorough. This work advances our understanding of Src mediated NMDAR regulation and the role of ND2 in the signaling, and will stimulate a considerable amount of new experiments. Below are listed several points that the authors should consider, which may strengthen the manuscript.

1. Throughout the paper, I wondered if the ND2-NMDAR interactions are influenced by receptors that function vs those that do not. I suspect many of the constructs as well as (of course) expression of a subunit in isolation will not be functional. Some confirmation that these mechanistic experiments did not provide false positive in terms of interactions, which may be qualitatively different for intact receptors. Some demonstration that various constructs do or do not make functional receptors on the surface of the cell would be helpful, I think, for the reader. In addition, a sentence or do pointing towards the data confirming results found with non-functional constructs had for functional constructs is warranted.
2. The data are fascinating, but also raise the question as to whether ND2 is present in all heterologous expression systems. If this is indeed an accessory subunit, as a mitochondrial protein it is presumably present in all commonly used heterologous expression systems (CHO cells, BHK cells, HEK cells, oocytes). Some discussion of whether the field has already been studying currents with this accessory subunit present (unaware of its presence) would be helpful. Alternatively, does ND2 need to be explicitly added to recapitulate the NDMAR-ND2 complex. Is ND2 present equally throughout all regions of the brain?
3. Some biochemistry confirming coassembly would be a very nice complement to the molecular/imaging studies.
4. Perhaps clarify what is meant by statement that the CTD as disordered? Parts are known to bind to intracellular proteins, presumably with tight order at the residues mediating the binding contact.
5. The authors might consider to discuss in more detail why ND2 might not have interactions with other TMD regions.
6. In Fig 6a (bar graph): error bars indicating SEM or STD range both ways (+/- of a central domain makes the range of error). However, the error bar of the first group implies negative intensity, which of course is not theoretically possible? Although a minor point, I'm not sure this is the correct way to present these data (intensity cannot theoretically be negative).
7. Although minor, it might be worth considering the use of different colors for these Cumulative Probability graphs (supplementary Fig 2) to help the reader grasp the result.

Reviewers' comments: (included in blue text, followed by our response):

Reviewer #1

We are pleased that Reviewer 1 judges our study to be “very interesting and carefully conducted” and that it “significantly advances the field”. We thank the Reviewer for his/her comments and in the revised manuscript have included key new experiments and analysis to address the Reviewer’s concerns. We trust that the Reviewer will concur that these have further improved the quality of the work.

Specific comments:

1) Analysis of the evolutionary conservation of the amino acids on both sides of the transmembrane interface, for example using ConSurf (<http://consurf.tau.ac.il/>), may further validate the molecular model. The anticipation is that the interface would be more highly conserved than the periphery. This approach has been used, for example, by Gofman et al 2012, Structure DOI: 10.1016/j.str.2012.05.016 (for examination of a different docking complex).

Response: We have now performed analysis of the evolutionary conservation of the amino acids on both ND2 and NMDAR using the ConSurf server (new Supplementary Fig. 7). From this analysis we found no evidence that either the ND2 groove or the exposed M4 surface of GluN1 is more highly conserved than the periphery. For example, TMs 1 and 11 of ND2, which line the outer rim of the ND2 groove, are among the least conserved helices while TMs 2, 3 and 10, which are not part of the groove, are the most highly conserved. For the TMD of the NMDAR, the M2 and M3 helices, which form the channel, are more conserved than both M1 and M4. Most of the residues that are highly conserved in M4 are forming intramolecular interactions with M1/M3 of the subsequent GluN2 subunit. These results suggest that the interacting surfaces from both ND2 and NMDAR are flexible. This is in agreement with our previous experimental observations, that substitution of GluN1 M4 for either the M4 of GluN2A or acetylcholine receptor α , maintained the ND2:GluN1 interaction (Fig. 4d). This new analysis fully supports the conclusions of our original manuscript, that it is structural complementarity, rather than amino acid specificity, that is the critical factor facilitating ND2:NMDAR interaction.

We have included a discussion of this analysis on page 13 of the revised manuscript and in the figure legend of Supplementary Fig. 7.

2) How about testing the effect of point mutations in key residues on the interaction? It would provide validation of the molecular details of the docking model, which is tentative otherwise. For example, the rotation angle of the M4 helix around its axis might be wrong. The ConSurf analysis above may highlight specific amino acids to mutate.

Response: The ConSurf analysis did not highlight specific amino acids to mutate except for the first six residues, “NMAGVF”, which are identical between GluN1 and GluN2. In order to address the dependence of the ND2:NMDAR interaction on the GluN M4 region, we created two new GluN constructs, both of which contained a key single amino acid change in GluN M4. In GluN1 subunits, a conserved and critical methionine immediately

follows the first six M4 residues, “NMAGVF” (See Ren *et al*, J. Biol. Chem. 278:276-283, 2003). However, GluN2 subunits have an additional tyrosine present prior to this methionine (Supplementary Table 1). We examined whether this key difference between GluN subtypes was a critical component governing the ND2:GluN1 interaction. We created a new GluN1 construct, GluN1^{insertY818}, in which a tyrosine residue was inserted directly prior to the critical methionine. The insertion of this tyrosine in GluN1 alters the helical register for all the residues C-terminal to this insertion, including the critical methionine, thereby changing which amino acid side chains are exposed to the ND2 groove. The PCC values for GluN1^{insertY818} with GFP-ND2 was not significantly different from that of GluN1 (new Supplementary Fig. 5b). As described in the original manuscript, more substantial modifications to the GluN1 M4, including complete and partial M4 substitutions, gave similar results, all displaying PCC values that did not differ significantly from that observed for GluN1 alone (Fig 4d and Supplementary Fig. 6a, c). Thus, these new data further substantiate our original conclusions that only a transmembrane M4 helix is needed, including those with varying primary amino acid sequence, to provide the structural complementarity required for GluN1 to bind to ND2 (new Supplementary Fig. 5b). We also created another new construct, GluN2A^{ΔY822}, in order to examine whether the deletion of this tyrosine in GluN2A M4, again resulting in a shift of the M4 helical register, was sufficient to facilitate colocalization of GluN2A with GFP-ND2. The PCC obtained for GluN2A^{ΔY822} was significantly lower than that of GluN1 or GluN1^{insertY818}, (Supplementary Fig. 5b), indicating that shifting the helical register of GluN2A M4 to more closely align with that of GluN1 was not sufficient to facilitate interaction between GluN2A and ND2. This result is in keeping with our previous observation that the replacement of the native GluN2A M4 with GluN1 M4, (construct GluN2A^{N1M4}), did not permit interaction between ND2 and GluN2A (Fig. 4d and Supplementary Fig. 6b). Importantly, we note that the additional colocalization results obtained with both of these new M4 helical register shift mutants (GluN1^{insertY818} and GluN2A^{ΔY822}), fully support the conclusions of our original manuscript.

Similarly, as reported in the original manuscript, all of the other GluN1 M4 substitutions, both full (GluN1^{AChr M4}), and partial, (GluN1^{A/N M4} and GluN1^{N/A M4}) also did not lead to a significant difference in GFP-ND2 colocalization PCC when compared to that obtained for full length GluN1 (Fig. 4d). The docking model is validated by the loss of interaction upon more dramatic removal of interacting helices. Based on the lack of distinct conservation in only GluN1 M4 and in the ND2 groove (See Response to Reviewer #1, comment 1) and the fact that these mutagenesis studies showed no perturbation of the interaction, including changing the helical register of GluN1 M4 with the tyrosine insertion in GluN1^{insertY818}, we came to the same conclusion as the Reviewer, that the exact sequence of the transmembrane region is of secondary importance, and thus “point mutations might have little or no effect on the interaction”.

We have clarified this last point on page 12 of the revised manuscript.

3) The results seem to suggest that the exact nature of the transmembrane region is of secondary importance, which is surprising in view of the specificity observed. If so, the anticipation is that the interface might be less evolutionarily conserved, and point mutations might have little or no effect on the interaction. It is worthwhile testing.

Response: Answer incorporated in Comment 2 to Reviewer #1.

4) Regarding the evolutionary scenario, do higher organisms carry also the long NMDAR variant, or was it completely lost in evolution?

Response: Higher organisms have completely lost the ability to express long variants of ND2 according to Birrell & Hirst, *J. FEBS Lett.* 584, 4247–4252, 2010. Unlike Cnidarians, which contain two in-frame initiation codons, the

first for the long and the second for the short ND2 variant, bilaterians contain an in-frame terminator codon just upstream of the second initiation codon. However, this does not rule out the possibility of higher organisms expressing the 'lost' first three N-terminal helices independently from the 'short' ND2. The presence of a functional 'short' ND2 was confirmed by the low-resolution EM structure bovine Complex 1 (Vinothkumar et al, *Nature* **515**, 80–84 (2014)).

We have clarified this point on page 7 of the revised manuscript.

5) The modeling procedure used makes sense, but alternatives should be examined as well. In particular:

(a) Other tools for modeling the ND2 structure should be used. Two options that are easily accessible are HHpred (<http://toolkit.tuebingen.mpg.de/hhpred>) and FFAS03 (<http://ffas.sanfordburnham.org/ffas-cgi/cgi/ffas.pl>).

(b) Similarly, also more tools should be used for docking. For example, PatchDock (<http://bioinfo3d.cs.tau.ac.il/PatchDock/>) and GRAMMX (<http://vakser.compbio.ku.edu/resources/gramm/grammx/>).

Response: Before embarking upon modelling human ND2, we tested various modelling programs including HHpred, ModWeb, M4T, SWISS-MODEL, I-TASSER, Phyre2 and IntFOLD2 using the webserver http://www.proteinmodelportal.org/?pid=modelling_interactive. With the availability of multiple high resolution crystal structures of NADH dehydrogenase (ND) subunits from various Complex 1 structures and the high secondary structural conservation between ND subunits from diverse species, these programs were picking the same structures of ND subunits as templates as Phyre2 and the models they produced were practically identical. However, Phyre2, which makes use of state-of-the-art approaches in terms of modeling, was chosen for all consequent analysis because it is among the best methods for protein structure prediction. It has been cited over 1000 times and has been ranked among the best systems of its kind for the last four years as judged by the biannual Critical Assessment of Structure Prediction (CASP) meetings. The server makes use of HHpred 1.51 to detect templates, Psi-Pred 2.5 to predict secondary structure, Disopred 2.4 to detect disordered regions, Memsat_SVM to predict transmembrane moieties, and Poing 1.0 for Multi-template modelling and ab initio. The complete model generated by Poing is combined with the original templates as input to MODELLER.

We initially tried using docking programs such as GRAMM-X and HADDOCK for the docking of ND2 and NMDAR. However, these docking programs failed to successfully dock ND2 onto the NMDAR mainly due to the conformational changes required from both the TMD of NMDAR and the ND2 to permit docking and prevent steric clashes. Because we did not have high resolution experimental data such as mutagenesis or NMR restraints required to guide these programs, we undertook a structural complementarity-guided docking approach. Our rationale was that if ND2 was a single domain membrane protein with very few long extra-membranous loops or tails, with a transmembrane groove, the most probable mechanism was a TMD:ND2 mechanism of interaction. The goal of our docking was to generate such a simple model that fits our vision of the system using the missing third helix from the long ND2 as our guide. Having this docked complex, we designed ND2 and NMDAR constructs to experimentally test and validate such a TMD:ND2 interaction.

6) After validation, the coordinates of the model structure should be made publically available to examine further tests, and usage, by others.

Response: As of now, we cannot deposit the model of this complex into any publicly available structure database since this is not yet an experimentally determined structure. However, the pdb co-ordinates and scripts used will be made available upon request. In the near future, we plan to collaborate with the X-ray crystallographers or cryo-electron microscopists with expertise in characterizing membrane protein complexes

to experimentally solve the structure of the NMDAR:ND2 complex. Although NMDAR and Complex I crystal structures have been solved by multiple groups, so far isolated ND2 is not yet amenable to high protein expression necessary for structure determination.

7) There are many abbreviations, which complicated my reading of the paper. Maybe some could be omitted. For example, PSD?

Response: We edited the manuscript to avoid using abbreviation whenever possible and have ensured that all abbreviations are defined when first used. We have altered “PSD” to its non-abbreviated form, “post-synaptic density”, throughout the revised manuscript.

8) I would add a figure of the whole complex in the main text to make it easier for the reader to follow the various interactions. Some is presented in Fig. 7, but it comes too late to my taste.

Response: While we placed the figure of the entire ternary complex in Figure 7, we have shown the binary complex between NMDAR and ND2 as early as in Figure 2 to make it easier for the reader to follow. We believe that placing Figure 7 earlier without discussion of how we arrived at that model could potentially confuse the reader.

Reviewer #2

We thank Reviewer 2 for his/her comments on our “bold approach” to what is “undoubtedly an important subject”. We appreciate the Reviewer’s suggestions to improve and clarify the paper, and in the revised manuscript have included key new experiments to address the points raised.

Specific comments:

1. Fluorescent microscopic method to probe localization of ND2 and NMDARs is a good start. This work requires some biochemical experiment (IgG-based pull-down assay) to confirm that direct interaction occurs. There has to be some correlation is with the crude microscopic observation and the actual protein-protein interaction. If ND2 and NMDARs are interacting as the authors suggested through modeling, ND2 should be pulled down convincingly by anti-GluN1 and anti-GluN2 antibodies. Alternatively, interaction of ND2 with purified NMDAR proteins may be a very convincing option.

Response: We concur with the Reviewer that an immuno-based pulldown approach would provide additional demonstration of the ND2:GluN1 interaction. We have previously demonstrated both co-immunoprecipitation of GluN1 by ND2, and co-immunoprecipitation of ND2 by GluN1 from post-synaptic density preparations (Gingrich et al, *PNAS*, 2004). However, co-immunoprecipitation is not an appropriate method by which to prove a direct interaction between two proteins; co-immunoprecipitation does not exclude the possibility of an indirect interaction. The colocalization assay approach taken in the present study enabled us to screen for both interacting proteins and protein regions. Subsequently, the BiFC method was used to establish that ND2 and GluN1 were interacting in a direct fashion. Based on our colocalization and BiFC assays, as well as our previous co-immunoprecipitation data, we have demonstrated that the ND2:GluN1 interaction is consistently observed by different methodologies.

2. Regarding data in Fig. 4 and 5, truncation of the transmembrane domains in GluN1 and ND2 is extreme. Do authors know that truncation of TMD4 in GluN1 expressed alone is properly folded? Or, in the case of ND2, do authors know that the proteins are even expressed?

Response: Both full length GFP-ND2 and GFP-ND2 fragments in transfected HEK cells were consistently observed by widefield microscopy. All ND2 constructs were GFP-chimeras, and GFP signal was routinely observed in every HEK cell population. In the revised manuscript we have clarified this in the Results and Methods sections. Additionally, we have generated new data demonstrating that GFP-ND2 fusion proteins of the predicted molecular weights can be detected by anti-GFP western blot (new Supplementary Fig. 10).

While truncation of the GluN1 M4 region may seem to be extreme, it has been reported previously that oocytes injected with both GluN1 Δ M4 Δ CTD and full length GluN2A constructs demonstrate comparable cell surface expression to that of full length GluN1 + GluN2A, (See Meddows et al., *J. Biol. Chem.* 276(22):18795-803, 2001). As surface expression of GluN2A requires correctly folded GluN1 in order to facilitate export we surmise that the GluN1 Δ M4 Δ CTD construct has the capacity to fold correctly. However, we were mindful of minimizing the potential for disruption where possible. For this reason we created a series of full length GluN constructs in which the M4 region was modified (also see comment 2 to Reviewer #1). This M4 region modification was either marginal, by the insertion or deletion of a single tyrosine, GluN1^{insertY818} and GluN2A ^{Δ Y822}; complete, whereby another transmembrane region was substituted into a GluN construct in place of the GluN M4 region, GluN2A^{N1M4}, GluN1^{N2AM4}, and GluN1^{AChr M4}; or partial, where half of the GluN1 M4 was substituted for half of the acetylcholine receptor α transmembrane region M4, GluN1^{A/N M4} and GluN1^{N/A M4}. In all of the GluN1 constructs,

the M4 modification did not lead to a significant difference in GFP-ND2 colocalization PCC when compared to that obtained for full length GluN1.

3. Could the result that TMD4-truncated GluN1 does not interact with ND2 caused by non-tetrameric assembly of this construct? In all of the experiments, the authors should co-express GluN2 subunit along with the GluN1 mutants since expression of GluN1 alone does not represent a physiological state.

Response: As requested, we have undertaken additional experiments where we carried out the colocalization assay with GluN1 truncation mutants + full length GluN2A + ND2. In each case, the inclusion of the GluN2A subunit did not alter our conclusions, and GluN2A transfected cells tested did not display significantly different PCC values from those that were not transfected with GluN2A (new Supplementary Fig. 4). We have described the PCC data obtained on page 10, paragraph 1 of the revised manuscript.

In addition to the GluN1 + GluN2A + full length GFP-ND2 data described in the original manuscript (Fig. 3d), we have carried out new experiments where we have examined the colocalization of GluN1 with four GFP-ND2 fragment constructs where GluN2A was also co-transfected. Notably, the inclusion of the GluN2A subunit did not alter our conclusions; GluN2A + GFP-ND2 fragment + GluN1 transfected cells did not give significantly different PCC values from those that were transfected with GFP-ND2 fragment + GluN1 only. We have included this new data as part of the supplementary materials (new Supplementary Fig. 9), and have described the PCC data obtained on both page 14, paragraph 1, and page 15, paragraph 1 of the revised manuscript.

4. As an alternative approach to truncation, can authors mutate GluN1-TMD4 to alter interaction with ND2? The authors may test the mutant constructs for the NMDAR activity and assess interaction with microscopy and pull-down.

Mutation of GluN1-M4 to alter interaction with ND2 is an interesting suggestion, and as requested we created new GluN M4 mutants and tested their interaction with ND2. We have described both the mutagenic strategy adopted and results obtained for our GluN M4 mutants in our response to Reviewer #1 (Comment 2). This additional data (new Supplementary Fig. 5b) is described in pages 14-15 of the revised manuscript.

-Fig.5a, More detail on membrane topology would help - the authors should show which side of the schematics is extracellular and cytoplasmic.

Response: Fig 5a has been amended as requested.

-Fig.6C. It appears that the decay time of aspartic acid induced current is significantly slower in ND2 TM6-8 compared to ND2 TM6-7 or GFP control. Could it possibly be the difference between ND2 associated NMDARs and non-associated NMDARs? Would the authors see the difference in decay time without Src activating peptides?

Response: Although the sample traces depicted in Fig. 6c do appear to show a difference in the decay times of aspartic acid induced current between ND2 TM 6-8 and both ND2 TM 6-7 and the control, this is due to the individual traces chosen, and was not indicative of the complete datasets. A comparison of the decay times observed for the ND2 TM 6-8 (n = 14) and 6-7 (n = 9) populations revealed that there was no significant difference in decay time between the two cohorts ($2215 \pm 321\text{ms}$ and $1734 \pm 292\text{ms}$ respectively; $p > 0.05$).

-Related to the above question, it would be worth conducting an experiment where NMDAR (for example, GluN1/GluN2A or B) is co-expressed with ND2 in HEK cells and deactivation and desensitization time course is

measured. There is a possibility that ND2 may act like the NMDAR-equivalent of TARP and Neto2 for AMPA and kainate receptors.

Response: This is a very interesting proposition, and one that we would be keen to investigate in a future study. In particular, we would be interested in the effect that expression of the key interacting region of ND2 may have on NMDAR deactivation and desensitization kinetics.

Reviewer #3

We are pleased that the Reviewer found our study “highly novel”, “significant and timely” and that the experimental designs were judged “clever and well-executed”. We appreciate his/her comments regarding our manuscript and the detailed suggestions. We have addressed the comments below, added data from additional experiments, and made clarifications as requested.

Specific comments:

1. Throughout the paper, I wondered if the ND2-NMDAR interactions are influenced by receptors that function vs those that do not. I suspect many of the constructs as well as (of course) expression of a subunit in isolation will not be functional. Some confirmation that these mechanistic experiments did not provide false positive in terms of interactions, which may be qualitatively different for intact receptors. Some demonstration that various constructs do or do not make functional receptors on the surface of the cell would be helpful, I think, for the reader. In addition, a sentence or do pointing towards the data confirming results found with non-functional constructs hod for functional constructs is warranted.

Response: Most of the PCC data reported in our original manuscript was with GluN1 alone, and as noted, this subunit alone is non-functional. To address the above point raised by Reviewer #3 we carried out additional experiments in which we co-expressed GluN2A with the GluN1 and ND2 constructs. In each case, the inclusion of the GluN2A subunit did not alter our conclusions; PCC values obtained were not significantly different between cells transfected with GluN1 + 2A constructs vs GluN1 alone (new Supplementary Fig. 4. 9). (We have further expanded on the data obtained in comment 3 to Reviewer #2). The cell surface expression and functionality of GluN1 Δ CTD, Δ ATD and Δ M4 mutants when co-expressed with GluN2A have been reported previously, with many of the GluN1 truncation mutants observed at the cell surface and functional (See Meddows et al., *J. Biol. Chem.* 276(22):18795-803, 2001).

2. The data are fascinating, but also raise the question as to whether ND2 is present in all heterologous expression systems. If this is indeed an accessory subunit, as a mitochondrial protein it is presumably present in all commonly used heterologous expression systems (CHO cells, BHK cells, HEK cells, oocytes). Some discussion of whether the field has already been studying currents with this accessory subunit present (unaware of its presence) would be helpful.

Response: As a core component of complex 1 in the mitochondria, ND2 is understood to be present in every functional mitochondrion. As demonstrated both in the current work and previously (Gingrich et al, *PNAS*, 2004 and Liu et al, *Nature Medicine*, 2008) ND2 is required for Src-mediated upregulation of NMDAR current. We would infer that ND2 is present in expression systems where this upregulation phenomenon is observed. We have commented on the possibility raised by the Reviewer on page 19 of the revised manuscript.

Alternatively, does ND2 need to be explicitly added to recapitulate the NDMAR-ND2 complex.

Response: Currently, the mechanism by which native ND2 is retained in mitochondria or exported to the plasma membrane of neurons to mediate the upregulation of NMDAR currents by c-Src is not fully understood. However, our electrophysiological work (Fig. 6) indicates that the addition of GFP-ND2-TM6-8, but not GFP-ND2-TM-6-7, was sufficient to disrupt the NMDAR:ND2:Src complex. This suggests that ND2 is present endogenously.

- Is ND2 present equally throughout all regions of the brain?

Response: Although we have not investigated the distribution of ND2 throughout all regions of the brain, we have previously shown that it is present at PSDs of the CNS, which are a prominent subcellular localization for c-Src. However, the Reviewer raises an interesting possibility, and one that we are keen to investigate in future studies.

3. Some biochemistry confirming coassembly would be a very nice complement to the molecular/imaging studies.

Response: As mentioned in our response to Reviewer #2 (comment 1), in our previous PNAS paper we have done biochemical experiments demonstrating both Co-IP and pulldown between ND2 and NMDAR subunits. Furthermore, when we abrogated the expression of mitochondrial proteins including ND2, we lost the Src-mediated enhancement of NMDAR activity. These biochemical studies confirmed that there is some interaction between ND2 and NMDAR, and in this current study we show that the interaction is direct through the GluN1 subunit.

4. Perhaps clarify what is meant by statement that the CTD as disordered? Parts are known to bind to intracellular proteins, presumably with tight order at the residues mediating the binding contact.

Response: We have clarified the meaning of CTD as disordered by adding the following statement on page 3 of the revised manuscript:

“Because of their amino acid compositions, intrinsically disordered proteins or protein regions such as the CTD of GluN2 of the NMDAR, lack stable secondary and tertiary structure, yet are increasingly recognized for the critical biological roles they play in mediating regulatory protein interactions, particularly those involving posttranslational modifications such as phosphorylation.” Interactions of binding motifs within disordered regions often give rise to only transient local order, not diminishing the overall disordered nature of the region.

5. The authors might consider to discuss in more detail why ND2 might not have interactions with other TMD regions.

Response: While the GluN1 M4 is the major interacting helix with the ND2 groove, TM1 and 11 of ND2 also interacts with the shallow TMD grooves formed M1 and M4, and M4 and M1 of the subsequent subunits. We have stated this previously (page 8) without describing what forms these shallow grooves. We have now clarified this further on page 7 of the revised manuscript as follows:

“Consequently, the M4 of each NMDAR subunit protrudes and exposes a significant amount of its surface, with two flanking shallow grooves formed between M1 and M4 of one subunit and between M4 of one subunit and M4 on the next subunit (Fig. 2). Prior to the computational docking simulation, we performed structural analysis using PyMol to investigate any geometric and/or physico-chemical complementarities between the two structures. For a direct ND2:TMD interaction, the protruding M4 of each GluN subunit seemed a strong candidate compared to M1 to fit in the ND2 groove.”

6. In Fig 6a (bar graph): error bars indicating SEM or STD range both ways (+/- of a central domain makes the range of error). However, the error bar of the first group implies negative intensity, which of course is not theoretically possible? Although a minor point, I'm not sure this is the correct way to present these data (intensity cannot theoretically be negative).

Response: We thank the Reviewer for his/her useful comments with regard to Figures 5f and 6a and have altered them accordingly.

7. Although minor, it might be worth considering the use of different colors for these Cumulative Probability graphs (supplementary Fig 2) to help the reader grasp the result.

Response: We have amended Supplementary Fig. 2 as requested.

Reviewers' Comments:

Reviewer #1 (Remarks to the Author)

The authors have addressed the comments well and I feel that the manuscript is ready for publication. Cheers!

Reviewer #2 (Remarks to the Author)

Identification of an auxiliary subunit for NMDAR, equivalent to TARP for AMPARs, is important. However, technical concerns regarding this work remains.

-Concluding protein-protein interaction with just a BiFC method is not sufficient to definitively state that NMDAR and ND2 are interacting. The authors need to show this definitively with purified proteins or blue native page. Immunoprecipitation was done by the authors in their paper published in 2004, but that should be done on mutants that they tested here.

-The major weakness of the work is that the authors do not validate proper folding and channel assembly of NMDARs. What does ND2-NMDAR interaction really mean if ND2 is not interacting with the NMDAR tetramer? The authors, at minimal, should show that the mutant channels (a series of M4 mutants) form ion channels by showing glutamate/glycine gated current that can be attenuated by specific inhibitors. For the M4 truncation or substitution (with a transmembrane domain from acetylcholine receptor) experiment, the authors need to show that the mutant NMDARs are assembled as tetramers, not aggregates. For example, the assessment by blue native PAGE to indicate proper oligomeric assembly is one possibility. Good examples are studies of AMPAR and interacting proteins as in Kim KS, Yan D, Tomita S. J Neurosci. 2010 Jan 20; 30(3):1064-72 and Gill MB, Kato AS, Roberts MF, Yu H, Wang H, Tomita S, Brecht DS. J Neurosci. 2011 May 4; 31(18):6928-38.

-Maintenance of ND-NMDAR interaction after substituting GluN1 M4 and the transmembrane domain of acetylcholine is rather surprising. The conclusion that this tight ND2-NMDAR interaction is mediated by the presence of a transmembrane helix at the position of GluN1 M4 is unordinary. Could the chimera NMDAR even be folded and assembled? Is it forming a channel? Even though the authors indicated this chimera has been used in the previous study by another group, validity of the approach in the paper is questionable. Again, concluding the interaction just by BiFC is too risky especially when results are a bit unordinary. After showing that this robust mutant is indeed forming a tetramer, the authors should at least show by immunoprecipitation that ND and NMDAR convincingly interact with each other.

-If the authors are not showing any functional alteration based on this interaction as for TARP in AMPARs and Neto for kainate receptors, the impact and significance of this work are low. In 2004, the authors have already shown that ND2-NMDAR interaction is necessary for Src phosphorylation of NMDARs. So, other than potentially delineating the interacting sites of the ND-NMDAR interaction, there are not many new pieces of information that this manuscript actually offers.

Reviewer #3 (Remarks to the Author)

The authors have done an excellent job at addressing the reviewers' concerns and should be congratulated on completion of an important and high quality study.

Response to Reviewer #2

In the review of the revised manuscript Reviewer 2 requested additional data to demonstrate further the GluN1:ND2 interaction, and to show that the receptors with the GluN1 mutants are functionally assembled. We have now done a number of further experiments in which we demonstrate that the ND2 TM6-8 fragment co-immunoprecipitates together with NMDARs, and that the receptors containing the GluN1 mutants are activated by NMDA with blockade by the NMDAR antagonist D-APV demonstrating functional assembly and trafficking to the cell surface. These experiments and findings are described below.

As requested, we have undertaken additional immunoprecipitation experiments in order to further validate the GluN1:ND2 interaction. In these experiments we used the ND2 fragment – ND2 TM6-8 – which our colocalization studies and bimolecular fluorescence complementation studies showed to interact with GluN1. We co-transfected HEK293 cells with both the GluN1 and GFP-ND2 TM 6-8 constructs, and 48 hours after transfection, immunoprecipitated the lysate with mouse anti-GluN1 antibody. Mouse IgG was used to control for non-specific pulldown. These IPs were then blotted using a rabbit anti-GFP antibody. We found that the GFP-ND2 TM 6-8 band of the expected size co-immunoprecipitated in the GluN1 immunoprecipitation; but GFP-ND2 TM 6-8 was not observed in the IgG pulldown. We have described this new data on page 15 of the revised manuscript and in Supplementary Fig. 12. We have also described the methods in the supplementary methods section, and in the figure legend of Supplementary Fig. 12. Thus, the immunoprecipitation data, as well as the colocalization and BiFC assays described in the current work, form a coherent data set demonstrating that the region comprising TM ND2 6-8 interacts with NMDARs.

In order to address the Reviewer's concerns about the functional assembly of NMDARs containing mutant GluN1 we have done a series of new electrophysiological experiments, as the Reviewer suggested that we "should show that the mutant channels (a series of M4 mutants) form ion channels by showing glutamate/glycine gated current that can be attenuated by specific inhibitors". Specifically, we expressed five GluN1 constructs, separately, together with GluN2A and PSD95 in HEK293 cells. The former consisted of one of: WT GluN1; a truncation mutant, GluN1 Δ ATD Δ M4 Δ CTD, or a GluN1 M4 substitution mutant – GluN1^{AChr M4}, GluN1^{A/N M4}, or GluN1^{N2AM4}. We made whole-cell patch clamp recordings 48 hours after transfection. With each of the GluN1 constructs we found that applying NMDA reproducibly evoked inward currents that were subsequently attenuated by D-APV, a specific antagonist of NMDARs. Our new electrophysiological findings provide robust evidence that the mutant receptors are correctly folded, tetrameric, functional NMDARs that are trafficked to the cell surface. Importantly, these experiments were done in the same HEK293 system in which we carried out our colocalization, BiFC and immunoprecipitation experiments. We appreciate the Reviewer's suggestion to do these new experiments. We have included a description of these additional data on page 13 of the revised manuscript and in Supplementary Fig 7. As well we have described the methods on pages 22, 23, 26, and in the figure legend of Supplementary Fig. 7.

We were pleased that Reviewer 2 appears to ascribe substantial novelty to our findings, describing them as "surprising" and "unordinary" on what he/she called an "undoubtedly an important subject". We too contend that there is considerable novelty and significance contained in our manuscript, a view that appears to be strongly held by the other Reviewers. Reviewer 1 judged our study to be "very interesting and carefully conducted", that it "significantly advances the field" and "is ready for publication".

Reviewer 3 previously described our study as “highly novel”, “significant and timely” and that the experimental designs were judged “clever and well-executed”. In the most recent communication Reviewer 3 states that “the authors have done an excellent job at addressing the Reviewers' concerns and should be congratulated on completion of an important and high quality study.”

As we have addressed all the comments with the new data we have provided, we trust that the Reviewer will concur with the other two Reviewers that the manuscript is ready for publication.

Reviewers' Comments:

Reviewer #2 (Remarks to the Author)

The authors responded to the majority of questions raised by conducting immunoprecipitation on the mutants and electrophysiology on some of the mutants they made. The authors chose not to answer the fundamental question of whether or not the ND2-NMDAR interaction affects the ion channel function, thus, the novelty of the work is limited to the fact that the the ND2-NMDAR interacts at the transmembrane domain in addition to the CTD shown previously by the authors.

Reviewer #4 (Remarks to the Author)

The authors have added data that help address reviewer concerns. Attention to several more issues could further strengthen the manuscript:

(1) p. 8, paragraph 2: The docked model is said to show that both GluN1 subunits may be able to interact simultaneously with a single ND2. This does not appear plausible based on images shown; please provide more explanation.

(2) Data on GluN1 M4 modification/truncation mutants: (a) Please provide information on the location in GluN1 of the M4 truncation. (b) Why was PSD95 co-expressed with NMDARs for the data shown in Supplementary Fig. 7? (c) Please provide the mean current amplitude across all cells from which recordings were made for Supplementary Fig. 7, and include information on cells that did not respond (if any). (d) Previous papers have suggested that the GluN1 M4 region is required for formation of functional channels, most directly Meddows et al. (2001; Table 1). Please address the apparent discrepancy with previous publications and conclusions concerning the importance of the GluN1 M4 region to receptor function.

(3) p. 17, paragraph 2: Should the reference to Fig. 6a instead be to 6b?

(4) p. 20, paragraph 1: It is stated that the ND2 interaction groove is exposed laterally in bacterial structures of Complex 1. Does this not conflict with a major point of the study, that the groove is blocked in bacterial homologs?

(5) p. 27, paragraph 2: In Methods use of Zipper-Vn and Zipper-Vc in both positive and negative controls is described; information on the results of the positive and negative controls should be included in Results.

(6) Fig. 2: I believe that the direction of image rotation shown in Fig. 2b and on the right in Fig. 2c are incorrect.

(7) Fig. 6 legend: What is the size of the scale bars in Fig. 6b?

(8) Supplementary Fig. 13: I believe there is a type in the legend: should "GFP-ND2-TM6-8 (b,c)" refer instead to GFP-ND2-TM6-7?

Response to Reviewer #4

In the review of the revised manuscript Reviewer 4 requested that we provide further discussion and clarify several points. We thank the Reviewer for his/her comments, which we believe have further improved the quality of the work. We have amended the manuscript as requested, and below include a point by point response to the comments.

(1) p. 8, paragraph 2: The docked model is said to show that both GluN1 subunits may be able to interact simultaneously with a single ND2. This does not appear plausible based on images shown; please provide more explanation.

We agree with the reviewer that a single ND2 can't interact with both GluN1 subunits. We meant that each GluN1 subunit can interact with one ND2. We have clarified this in the last sentence in paragraph 1 on page 8 of our revised manuscript.

(2) Data on GluN1 M4 modification/truncation mutants:

(a) Please provide information on the location in GluN1 of the M4 truncation.

Residue 813 of GluN1 is replaced by a stop codon in the GluN1 M4 truncation construct. This is now reported in the Methods section on page 23 (line 2) of the revised manuscript.

(b) Why was PSD95 co-expressed with NMDARs for the data shown in Supplementary Fig. 7?

Several groups (eg. Yamada *et al.*, 1999; Iwamoto *et al.*, 2004; Zukin *et al.*, 2004), have reported that PSD95 increases NMDAR currents in heterologous systems. We therefore included PSD95 in our transfections to increase the likelihood of observing NMDAR currents for the GluN1 M4 mutants. We have noted inclusion of PSD95 in the Methods section on page 22 (lines 6-8) of the revised manuscript, and in the legend of Supplementary Figure 7.

(c) Please provide the mean current amplitude across all cells from which recordings were made for Supplementary Fig. 7, and include information on cells that did not respond (if any).

As requested, we have now included the means for all cells tested in the figure legend of Supplemental Figure 7. Under the conditions reported all cells responded, and in every case tested current was blocked by APV.

(d) Previous papers have suggested that the GluN1 M4 region is required for formation of functional channels, most directly Meddows et al. (2001; Table 1). Please address the apparent discrepancy with previous publications and conclusions concerning the importance of the GluN1 M4 region to receptor function.

With regard to the function of NMDARs comprised of M4-lacking GluN1, we note that the currents we observed were approximately 60 pA on average, that is to say about 1/75th the amplitude of currents generated by receptors comprised of wildtype GluN1. The currents we observed are bona fide NMDA currents as they were fully blocked by the competitive NMDAR antagonist APV. As described above, our recordings of these currents were done with co-expression of PSD95. In contrast, Meddows et al. did not co-express PSD95. PSD95 co-expression is reported to increase NMDAR currents approximately 2-3 fold depending on heterologous cell type (eg. Zukin et al., J. Neurosci 2004) by increasing cell surface levels of NMDARs and by increasing channel opening rate. Thus, the lack of PSD95 co-expression NMDAR currents is the likely explanation of why Meddows et al. did not detect NMDAR currents. That NMDARs comprised of M4-lacking GluN1 can generate functional receptors is consistent with observations that such receptors bind glycine (Sandhu et al, J. Neurochemistry 1999); they can co-IP with GluN2A (Meddows et al, 2001); they exist as high molecular weight complexes

similar to those with WT GluN1 (Meddows et al, 2001); and they be trafficked to the cell surface at low levels (Meddows et al, 2001; Horak et al, 2008). We have included this explanation in the revised manuscript. Given that this is an ancillary finding in the present study, and the space restrictions on the main text, we have included this description in the legend of Supplementary Figure 7, where the recordings from these experiments are shown.

We find our observation on M4-lacking GluN1 intriguing and worthy of further investigation. However, we consider such investigation beyond the scope of the present study.

(3) p. 17, paragraph 2: Should the reference to Fig. 6a instead be to 6b?

Yes. This has been corrected.

(4) p. 20, paragraph 1: It is stated that the ND2 interaction groove is exposed laterally in bacterial structures of Complex 1. Does this not conflict with a major point of the study, that the groove is blocked in bacterial homologs?

We thank the Reviewer for pointing out confusing wording in this sentence - the groove is indeed blocked in the bacterial homologs. If the three N-terminal helices of bacterial ND2 were removed from the Complex I crystal structure, the ND2 groove would be exposed laterally similar to the bovine structure. We have clarified this in the revised manuscript by replacing the sentence with "Importantly, in the bacterial Complex 1 the ND2 groove is occluded by its three N-terminal helices, but in the bovine structure the absence of these three helices exposes the ND2 interaction groove laterally, and thereby permits binding to GluN1." (Amended sentence is on page 19, lines 13-15).

(5) p. 27, paragraph 2: In Methods use of Zipper-Vn and Zipper-Vc in both positive and negative controls is described; information on the results of the positive and negative controls should be included in Results.

Zipper-Vn and Zipper-Vc were used as controls to establish data acquisition parameters for the BiFC technique. Co-transfection of Zipper-Vn and Zipper-Vc was used to establish whether Venus fluorescence could be consistently observed. Co-transfection of Zipper-Vn and GluN1 Δ CTD-Vc was used to establish the signal/noise threshold for non-interacting proteins.

We have clarified our sentence on page 26 (lines 6-8) of the revised manuscript where we have included the following; "HEK cells were transfected with Zipper-Vn and Zipper-Vc to optimise the maximum for Venus signal measurement. HEK cells were transfected with GluN1 Δ CTD-Vc and Zipper-Vn to determine background level of cell fluorescence."

(6) Fig. 2: I believe that the direction of image rotation shown in Fig. 2b and on the right in Fig. 2c are incorrect.

We have corrected the direction of rotation in an updated Figure 2.

(7) Fig. 6 legend: What is the size of the scale bars in Fig. 6b?

These scale bars are 10 μ m. This has been corrected in the legend.

(8) Supplementary Fig. 13: I believe there is a type in the legend: should "GFP-ND2-TM6-8 (b,c)" refer instead to GFP-ND2-TM6-7?

This has been corrected in the legend.